# Interplay between PML NBs and HIRA for H3.3 dynamics following type I interferon stimulus

Constance Kleijwegt[1†], Florent Bressac[1], Coline Seurre[1], Wilhelm Bouchereau[1], Camille Cohen[1‡], Pascale Texier[1], Thomas Simonet[2], Laurent Schaeffer[2], Patrick Lomonte[1*], Armelle Corpet[1*]

[1]University of Lyon, Université Claude Bernard Lyon 1, CNRS UMR 5261, INSERM U 1315, LabEx DEVweCAN, Institut NeuroMyoGène (INMG), Pathophysiology and Genetics of the Neuron and Muscle (PGNM) laboratory, team Chromatin Dynamics, Nuclear Domains, Virus, Lyon, France; [2]Univ Lyon, Université Claude Bernard Lyon 1, CNRS UMR 5310, INSERM U 1217, Institut NeuroMyoGène (INMG), team Nerve-Muscle interactions, Lyon, France

**\*For correspondence:**
patrick.lomonte@univ-lyon1.fr (PL);
armelle.corpet@univ-lyon1.fr (AC)

**Present address:** [†]Institute of Human Genetics, UMR9002 CNRS-Université de Montpellier, Laboratory of Biology of Repetitive Sequences, Montpellier, France; [‡]Univ Montpellier, CNRS UMR 5235, Laboratory of Pathogen Host Interactions, Montpellier, France

**Competing interest:** The authors declare that no competing interests exist.

**Abstract** Promyelocytic leukemia Nuclear Bodies (PML NBs) are nuclear membrane-less organelles physically associated with chromatin underscoring their crucial role in genome function. The H3.3 histone chaperone complex HIRA accumulates in PML NBs upon senescence, viral infection or IFN-I treatment in primary cells. Yet, the molecular mechanisms of this partitioning and its function in regulating histone dynamics have remained elusive. By using specific approaches, we identify intermolecular SUMO-SIM interactions as an essential mechanism for HIRA recruitment in PML NBs. Hence, we describe a role of PML NBs as nuclear depot centers to regulate HIRA distribution in the nucleus, dependent both on SP100 and DAXX/H3.3 levels. Upon IFN-I stimulation, PML is required for interferon-stimulated genes (ISGs) transcription and PML NBs become juxtaposed to ISGs loci at late time points of IFN-I treatment. HIRA and PML are necessary for the prolonged H3.3 deposition at the transcriptional end sites of ISGs, well beyond the peak of transcription. Though, HIRA accumulation in PML NBs is dispensable for H3.3 deposition on ISGs. We thus uncover a dual function for PML/PML NBs, as buffering centers modulating the nuclear distribution of HIRA, and as chromosomal hubs regulating ISGs transcription and thus HIRA-mediated H3.3 deposition at ISGs upon inflammatory response.

## Editor's evaluation

This study nicely dissects unexpected crosstalk between PML nuclear bodies and the HIRA member of the H3.3 histone chaperone complex upon inflammatory stress. The work raises interesting perspectives on how availability of HIRA could be regulated by PML Nuclear Bodies for histone deposition, and transcriptional regulation of key interferon-stimulated genes. Overall, this is an important step forward in unveiling which epigenetic pathways might regulate immune responses.

## Introduction

Promyelocytic Leukemia Nuclear Bodies (PML NBs) are membrane-less organelles (MLOs), also called biomolecular condensates (*Banani et al., 2017*), that concentrate proteins at discrete sites within the nucleoplasm thus participating in the spatio-temporal control of biochemical reactions (*Corpet et al., 2020*; *Hirose et al., 2022*; *Lallemand-Breitenbach and de Thé, 2018*; *Li et al., 2020*). PML

NBs are 0.1–1 μm diameter hollow sphere structures that vary in size and number depending on cell type, cell-cycle phase, or physiological state, highlighting their stress-responsive nature. The tumor-suppressor PML protein is the primary scaffold of PML NBs and forms an outer shell, together with the SP100 nuclear antigen, surrounding an inner core of dozens of proteins that localize constitutively or transiently in PML NBs. PML (also known as TRIM19) is a member of the tripartite motif (TRIM)-containing protein superfamily characterized by a conserved N-terminal RBCC motif essential for PML polymerization. Several isoforms of PML exist, all containing the RBCC motif and three well-characterized small-ubiquitin-related modifier (SUMO) modification sites at lysines K65, K160, and K490 and a SUMO interacting motif (SIM) enabling its interaction with SUMOylated proteins.

PML NBs have been involved in a wide variety of biological processes such as senescence, antiviral response, transcriptional regulation, DNA damage response, or stemness suggesting that they are fully significant structures. The molecular mechanisms through which they exert their broad physiological impact are not fully elucidated yet. However, as MLOs, they potentially have three different mode of actions (for review *Corpet et al., 2020*; *Hirose et al., 2022*). First, they can serve as hotspots promoting specific biochemical reactions such as SUMOylation. PML NBs are enriched in the SUMO E2 conjugating enzyme UBC9, which mediates SUMOylation of various proteins including PML thus enforcing PML-PML interactions via intermolecular SUMO-SIM interactions. SUMOylated proteins within PML NBs can drive the multivalent recruitment of inner core protein clients through their SIM, possibly via liquid-liquid phase separation (LLPS) mechanisms (*Corpet et al., 2020*; *Li et al., 2020*; *Sahin et al., 2014*). Second, PML NBs may buffer/sequestrate specific factors which could lead to their substantial depletion for the surrounding nuclear space. Third, while PML NBs are in general devoid of DNA, except in specific cases (for review *Corpet et al., 2020*), they reside in the interchromatin nuclear space (*Boisvert et al., 2000*) and can associate with specific genomic loci thus acting as chromosomal hubs adequately regulating associated genes (*Chang et al., 2013*; *Ching et al., 2013*; *Delbarre et al., 2017*; *Kumar et al., 2007*; *Kurihara et al., 2020*; *Shiels et al., 2001*; *Wang et al., 2004*). PML NBs have been found associated with both transcriptionally-active domains (*Boisvert et al., 2000*; *Kurihara et al., 2020*; *Wang et al., 2004*), as well as heterochromatin regions such as telomeres suggesting an important function in chromatin domain organization and regulation of their transcriptional state (for review *Delbarre and Janicki, 2021*).

Targeted deposition of histones variants is crucial for chromatin homeostasis and the maintenance of cell identity (*Allis and Jenuwein, 2016*). Among histone H3 variants, H3.3 is expressed throughout the cell cycle and is incorporated onto DNA in a DNA-synthesis-independent manner by dedicated histone chaperone complexes (*Martire and Banaszynski, 2020*). Histone cell cycle regulator A (HIRA) chaperone complex, composed of HIRA, ubinuclein 1 or ubinuclein 2 (UBN1 or UBN2) and calcineurin-binding protein CABIN1, is responsible for H3.3 deposition in transcriptionally active regions including enhancers, promoters and gene bodies, as well as in nucleosome-free regions and DNA damage sites (*Goldberg et al., 2010*; *Ray-Gallet et al., 2011*; *Ray-Gallet et al., 2002*; *Zhang et al., 2017*) (for review *Martire and Banaszynski, 2020*; *Ricketts and Marmorstein, 2017*). HIRA has also been recently shown to be involved in the transcription-mediated recycling of parental H3.3 by microscopy approaches (*Torné et al., 2020*). Although HIRA complex is diffusively distributed in the nuclei of proliferating somatic cells, it accumulates in PML NBs upon various stresses such as senescence entry (*Banumathy et al., 2009*; *Jiang et al., 2011*; *Rai et al., 2011*; *Zhang et al., 2005*), viral infection (*Cohen et al., 2018*; *McFarlane et al., 2019*; *Rai et al., 2017*), or interferon type I (IFN-I) treatment (*McFarlane et al., 2019*; *Rai et al., 2017*), which encompasses IFNα and IFNβ cytokines. These latter events underscore a role of HIRA in intrinsic anti-viral defense via chromatinization of incoming viral genomes (*Cohen et al., 2018*; *Rai et al., 2017*) as well as stimulation of innate immune defenses in the case of viral infection (*McFarlane et al., 2019*).

However, the exact significance of HIRA localization in PML NBs upon inflammatory stress response, as well as the role of the PML NBs themselves, remain to be defined. PML NBs may act as concentrating places for enzymatic reactions such as SUMOylation, or as buffering/sequestration structures for various chromatin-related proteins, or be a means to target them to specific chromatin regions juxtaposed to PML NBs. Here, we explored the different functions of PML NBs in response to IFN-I stimulus and focused our investigation on their interplay with the histone chaperone HIRA. We show that PML NBs act as nuclear depot centers for HIRA, depending on PML/SP100 levels, as well as on DAXX/H3.3 histone availability. Mechanistically, HIRA localizes in PML NBs in a SP100 and

SIM-SUMO-dependent manner upon IFN-I treatment. In addition, we provide evidence that PML is required for interferon-stimulated genes (ISGs) expression, and that ISGs loci juxtapose to PML NBs, thus confirming a role of PML NBs as chromosomal hubs implicated in gene regulation. ChIP-Seq analysis reveals a long-lasting H3.3 deposition on the 3' end of ISGs, which is partly dependent on HIRA and PML, but independent of HIRA localization in PML NBs. Together, our results put forward a dual role of PML/PML NBs during the inflammatory response: they act both as buffering centers to modulate HIRA availability in the nucleoplasm, and as chromosomal hubs regulating ISGs transcription, and thus, HIRA-mediated H3.3 deposition at ISGs.

## Results

### HIRA accumulation in PML NBs correlates with an increase in PML and SP100 concentration

Several stimuli, such as IFN-I treatment (*McFarlane et al., 2019*; *Rai et al., 2017*), can trigger HIRA accumulation in PML NBs. As a first step towards deciphering the mechanism of HIRA localization in PML NBs, we investigated the stoichiometry-dependent recruitment of HIRA in PML NBs. Treatment of human primary foreskin diploid fibroblast BJ cells with the TLR3 ligand poly(I:C), a strong stimulant of the IFN-I pathway, or with the Tumor necrosis factor α (TNFα) cytokine, triggered a strong accumulation of HIRA in PML NBs (*Figure 1A–B*), similarly to a control IFN-I treatment (*Figure 1—figure supplement 1A*). The accumulation was abrogated by addition of ruxolitinib, an inhibitor of the JAK-STAT pathway downstream of the IFN-I receptor, underscoring the involvement of the IFN-I signaling pathway in primary cells (*Figure 1A–B* and *Figure 1—figure supplement 1A*). IFNβ, poly(I:C) and TNFα induced an IFN-I dependent increase of the PML and SP100 proteins and their SUMOylated forms (*Figure 1C–D*), as well as their mRNA levels (*Figure 1—figure supplement 1B*) confirming previous data (*Gao et al., 2008*; *Guldner et al., 1992*; *Stadler et al., 1995*). Of note, IFNβ slightly increased HIRA mRNA (1.26 fold) and protein levels (1.34 fold) (*Figure 1—figure supplement 1C*). Treatment with other pro-inflammatory cytokines such as IL-6 or the IL-8 chemokine, did not increase PML protein levels or SUMOylation (*Figure 1—figure supplement 1D*), nor affected HIRA localization that remained pan-nuclear (*Figure 1—figure supplement 1E*). HIRA accumulation in PML NBs was not a consequence of senescence as cells treated with IFNβ continued to replicate as shown by EdU incorporation (*Figure 1—figure supplement 1F*). These results suggest that an IFN-I-dependent increase of PML and SP100 protein levels and SUMOylation, and to a lesser extent of HIRA, is part of the mechanism for HIRA accumulation in PML NBs.

### Accumulation of HIRA in PML NBs depends on both SUMO-SIM interactions and SP100

Interaction of a client SIM with SUMOylated lysines on the PML protein is critical for the recruitment of several PML NB clients such as SP100 (*Szostecki et al., 1990*), or the H3.3 histone chaperone DAXX (*Sahin et al., 2014*). The latter localizes constitutively in PML NBs (*Ishov et al., 1999*). Further studies suggest that the partitioning of polySIM chimeras in the condensed PML NB phase is controlled by the availability of SUMOylated lysines on the PML protein (*Banani et al., 2016*). We thus hypothesized that HIRA's partitioning in PML NBs could be regulated by SUMO-SIM interactions. We first investigated whether HIRA and PML/SUMO could interact together in cellulo. Proximity Labelling Assay (PLA) allows the detection of closely interacting protein partners in situ at distances below 40 nm (*Sahin et al., 2016*). Using PLA, we detected interaction foci between PML and SUMO2/3 as expected (*Sahin et al., 2016*), with the number of interaction foci increasing significantly upon IFNβ treatment (*Figure 2A–B*), which is known to stimulate PML SUMOylation (*Stadler et al., 1995*). We then assessed the interactions between HIRA and PML or HIRA and SUMO2/3. We could detect a significant interaction between these proteins in presence of IFNβ (*Figure 2A–B*), accordingly to the accumulation of HIRA in PML NBs. Positive PLA signal between HIRA and SUMO2/3 could either mean that HIRA is SUMOylated or that HIRA interacts with other SUMOylated proteins. The molecular mass of HIRA remained unchanged upon IFNβ treatment of primary cells (*Figure 1—figure supplement 1C*), as previously described (*McFarlane et al., 2019*), which is not in favor of post-translational modification of HIRA with SUMO groups. In addition, ectopic HIRA mutated on K809, identified as a possible SUMOylated lysine in a SUMO screen (*Hendriks et al., 2014*; *Schimmel et al., 2014*),

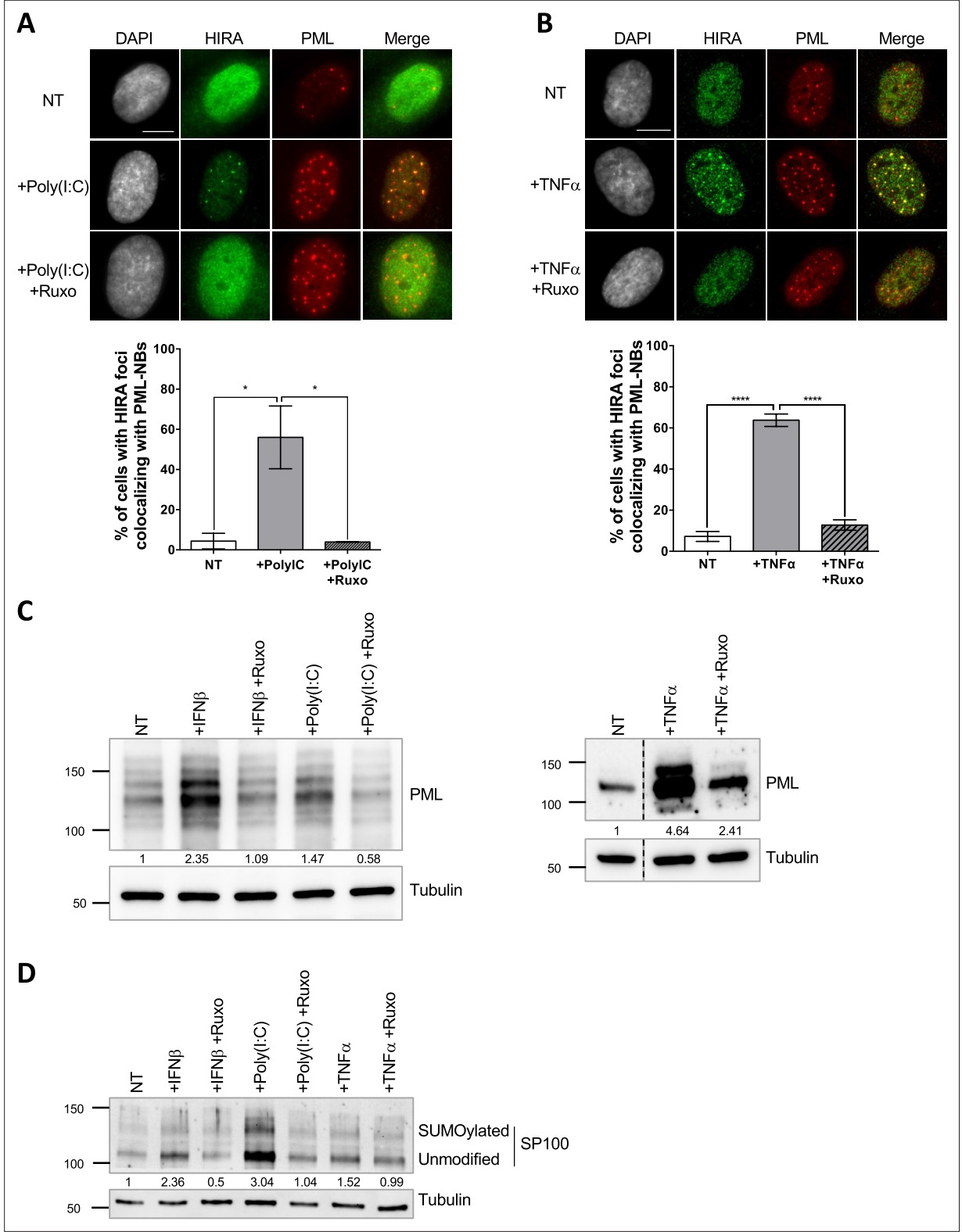

**Figure 1.** HIRA accumulation in PML NBs correlates with increased PML concentration in primary cells. (**A–B**) (top) Fluorescence microscopy visualization of HIRA (green) and PML (red) in BJ cells treated with Poly(I:C) at 10 µg/mL for 24 hr (left) or with TNFα at 100 ng/mL for 24 hr (right). Ruxolitinib (Ruxo) was added at 2 µM one hour before Poly(I:C) or TNFα treatment and for 25 hr. Cell nuclei are visualized by DAPI staining (grey). Scale bars represent 10 µm. (bottom) Histograms show quantitative analysis of cells with HIRA localization at PML NBs. p-values (Student t-test): *<0.05; ****<0.0001. Numbers on all histograms represent the mean of three independent experiments (± SD). (**C**) (left) Western blot visualization of PML from total cell

*Figure 1 continued on next page*

*Figure 1 continued*

extracts of BJ cells treated with IFNβ at 1000 U/mL or Poly(I:C) at 10 μg/mL for 24 hr and with ruxolitinib (Ruxo) at 2 μM 1 hr before treatment and for 25 hr. (right) Western blot visualization of PML from RIPA extracts of BJ cells treated with TNFα at 100 ng/mL for 24 hr and with ruxolitinib (Ruxo) at 2 μM 1 hr before treatment and for 25 hr. Tubulin is a loading control. Quantification of PML levels relative to tubulin are shown below the WB (numbers are representative from three independent experiments) (**D**) Western blot visualization of SP100 from total cell extracts of BJ cells treated as in C. Quantification of SP100 levels relative to tubulin are shown below the WB (numbers are representative from two independent experiments).

The online version of this article includes the following source data and figure supplement(s) for figure 1:

**Source data 1.** Raw WB for *Figure 1C* (left panel) for PML and tubulin.

**Source data 2.** Raw WB for *Figure 1C* (left panel) for PML and tubulin with labels.

**Source data 3.** Raw WB for *Figure 1C* (right panel) for PML.

**Source data 4.** Raw WB for *Figure 1C* (right panel) for PML with labels.

**Source data 5.** Raw WB for *Figure 1C* (right panel) for tubulin.

**Source data 6.** Raw WB for *Figure 1C* (right panel) for tubulin with labels.

**Source data 7.** Raw WB for *Figure 1D* for SP100.

**Source data 8.** Raw WB for *Figure 1D* for SP100 with labels.

**Source data 9.** Raw WB for *Figure 1D* for tubulin.

**Source data 10.** Raw WB for *Figure 1D* for tubulin with labels.

**Figure supplement 1.** Impact of various cytokines on HIRA localization in PML NBs.

**Figure supplement 1—source data 1.** Raw WB for *Figure 1—figure supplement 1C* for HIRA.

**Figure supplement 1—source data 2.** Raw WB for *Figure 1—figure supplement 1C* for HIRA with labels.

**Figure supplement 1—source data 3.** Raw WB for *Figure 1—figure supplement 1C* for tubulin.

**Figure supplement 1—source data 4.** Raw WB for *Figure 1—figure supplement 1C* for tubulin with labels.

**Figure supplement 1—source data 5.** Raw WB for *Figure 1—figure supplement 1D* for PML.

**Figure supplement 1—source data 6.** Raw WB for *Figure 1—figure supplement 1D* for PML with labels.

**Figure supplement 1—source data 7.** Raw WB for *Figure 1—figure supplement 1D* for tubulin.

**Figure supplement 1—source data 8.** Raw WB for *Figure 1—figure supplement 1D* for tubulin with labels.

was still recruited in PML NBs similar to the wild-type protein (*Figure 2—figure supplement 1A*), suggesting that at least K809 SUMOylation is dispensable for HIRA recruitment in PML NBs. We thus conclude that HIRA can interact with SUMOylated proteins in situ.

To confirm that SUMOylation of cellular proteins, including PML, is required for HIRA partitioning in PML NBs, we depleted the pool of SUMO1/2/3 by siRNA treatment (*Figure 2C*). Depletion of SUMOs led to a significant decrease of HIRA accumulation in PML NBs upon IFNβ treatment (*Figure 2D*). Of note, in absence of SUMOs, PML NBs appear as large aggregates devoid of DAXX (*Figure 2—figure supplement 1B*), reminiscent of the alternative PML NBs structures observed during mitosis, in human embryonic stem cells or in human sensory neurons (*Corpet et al., 2020*). Thus, presence of SUMO proteins, that can undergo LLPS in vitro (*Banani et al., 2016*), seems key to promote partitioning of HIRA. Increasing the pool of free SUMOs by ectopic expression did not trigger HIRA accumulation in PML NBs (*Figure 2—figure supplement 1C*) suggesting that SUMOs need to be conjugated to specific proteins to trigger HIRA partitioning in PML NBs.

To further substantiate the requirements for non-covalent SUMO-SIM interactions in mediating HIRA accumulation in PML NBs, we used the Affimer technology, previously known as Adhiron. Affimers are artificial protein aptamers consisting of a scaffold with two variable peptide presentation loops that can specifically bind with high affinity and high specificity to their binding partners. A recent screen identified several Affimers that inhibit SUMO-dependent protein-protein interactions mediated by SIM motifs (*Hughes et al., 2017*). We selected the S1S2D5 Affimer that specifically targets both SUMO1 and SUMO2/3-mediated interactions and which possesses a consensus SIM motif (*Hughes et al., 2017*). Inducibly expressed S1S2D5-His Affimer showed a nuclear staining with accumulation of the Affimer in PML NBs (*Figure 3A–B*), as expected for a synthetic peptide that exhibits a SIM domain (*Banani et al., 2016*; *Hughes et al., 2017*). S1S2D5-His Affimer expression prevented the accumulation of HIRA in PML NBs upon IFNβ treatment (*Figure 3C*), without affecting HIRA nor PML protein levels (*Figure 3—figure supplement 1A*). Collectively, our results demonstrate

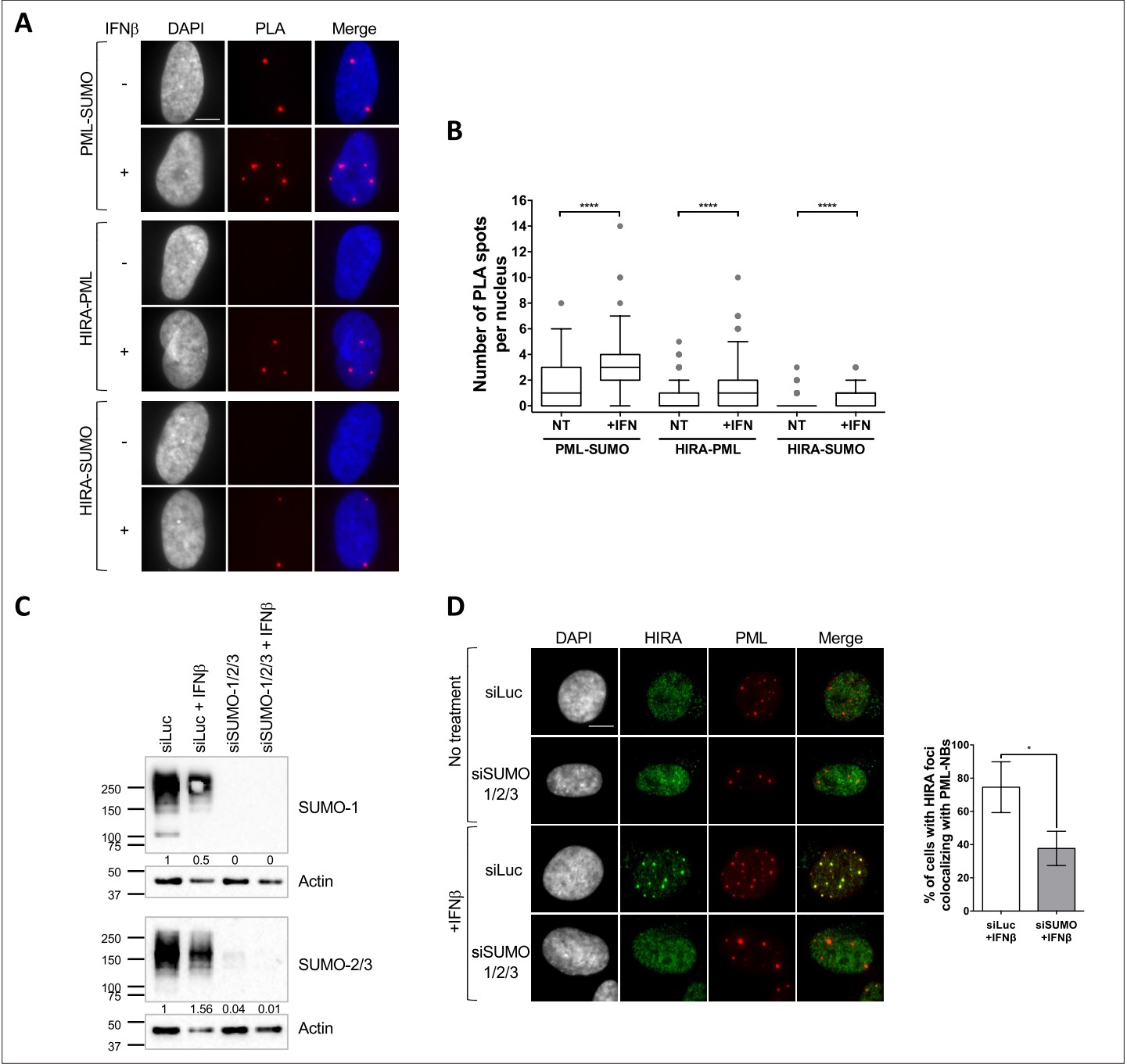

**Figure 2.** HIRA recruitment to PML NBs is dependent on SUMO proteins. (**A**) Fluorescence microscopy visualization of Proximity Ligation Assays (PLA) signals (red) obtained after incubation of anti-PML +anti SUMO, anti-HIRA +anti PML or anti-HIRA +anti SUMO antibodies on BJ cells treated or not with IFNβ at 1000 U/mL for 24 h. Cell nuclei are visualized by DAPI staining (grey or blue on the merge). Scale bar represents 10 µm. (**B**) Box-and-whisker plot shows the number of PLA spots detected in cells described in A. In average, 200 nuclei/condition were analyzed from three independent experiments. The line inside the box represents the median of all observations with interquartile range. p-values (Mann-Whitney u-test): ****<0.0001. (**C**) Western-blot visualization of SUMO-1 and SUMO-2/3 from total cellular extracts of BJ cells treated with the indicated siRNAs for 48 hr and with IFNβ at 1000 U/mL during the last 24 hr. Actin is a loading control. Quantification of SUMO-1 and SUMO-2/3 levels relative to tubulin are shown below the WB (numbers are representative from three independent experiments). (**D**) (left) Fluorescence microscopy visualization of HIRA (green) and PML (red) in BJ cells treated with siRNAs as described in C. Cell nuclei are visualized by DAPI staining (grey). Scale bar represents 10 µm. (right) Histograms show quantitative analysis of cells with HIRA localization at PML NBs. Numbers represent the mean of three independent experiments (± SD). p-value (Student t-test): *<0.05.

The online version of this article includes the following source data and figure supplement(s) for figure 2:

*Figure 2 continued on next page*

*Figure 2 continued*
**Source data 1.** Raw WB for *Figure 2C* for SUMO1.
**Source data 2.** Raw WB for *Figure 2C* for SUMO1 with labels.
**Source data 3.** Raw WB for *Figure 2C* for Actin.
**Source data 4.** Raw WB for *Figure 2C* for Actin with labels.
**Source data 5.** Raw WB for *Figure 2C* for SUMO2/3.
**Source data 6.** Raw WB for *Figure 2C* for SUMO1 with labels.
**Source data 7.** Raw WB for *Figure 2C* for Actin (bottom panel).
**Source data 8.** Raw WB for *Figure 2C* for Actin (bottom panel) with labels.
**Figure supplement 1.** HIRA accumulation in PML NBs does not depend on overexpression of SUMO proteins nor on its SUMOylation.

that SUMO-SIM interactions play an important role in the targeting of HIRA in PML NBs in response to IFNβ.

PML is known to be mainly SUMOylated on lysines K65, K160, and K490 (*Kamitani et al., 1998*). Immortalized *Pml^-/-* mouse embryonic fibroblasts (MEFs) reconstituted with a doxycyclin-inducible wild-type Myc-tagged version of human PML (Myc-PML WT) or a PML mutated on its three main SUMOylation sites (Myc-PML 3K) were used to investigate the specific requirements for PML SUMOylation in HIRA partitioning (*Figure 3D*). We first verified that HIRA accumulation in PML NBs was conserved in wild-type but not *Pml^-/-* MEFs upon activation of the IFN-I pathway (*Figure 3—figure supplement 1B*). Upon doxycyclin induction, Myc-PML WT or its mutated form were expressed at high levels in *Pml^-/-* MEFs (*Figure 3D*). Despite a recurrent diminution in the amount of the ectopic PML proteins following addition of mouse IFNα, possibly because of a lower efficiency of the doxycyline induction of ectopic PML transcription in IFN-I treated cells (*Figure 3D*), the wild type PML rescued HIRA accumulation in ectopically formed PML NBs unlike the PML 3K, which showed no colocalization with HIRA (*Figure 3E*). Of note, super resolution microscopy analyses of PML 3K-expressing MEFs reveal that PML 3K form spherical structures exactly like WT PML (*Sahin et al., 2014*). These data demonstrate that PML SUMOylation on K65, K160, and K490 is required for HIRA recruitment in PML NBs.

Multivalent interactions between client SIM motifs and SUMOylated lysines on the PML protein are implicated in client recruitment in PML NBs, as shown for DAXX (*Banani et al., 2016*; *Sahin et al., 2014*). Using JASSA (*Beauclair et al., 2015*) and GPS-SUMO (*Zhao et al., 2014*), we selected a set of five putative SIM motifs in HIRA protein sequence and tested whether they were involved in HIRA recruitment in PML NBs by mutating them individually (*Figure 3F*). Cells expressing the wild-type (WT) tagged version of HIRA (HIRA-HA WT) displayed ectopic HIRA accumulation in PML NBs upon IFNβ treatment (*Figure 3F*), as well as without IFNβ treatment (*Figure 4A* and *Figure 3—figure supplement 1C*). HIRA-HA mSIM1 and mSIM3 mutants did not show sufficient expression in individual cells to analyze their localization. HIRA-HA mSIM4 and mSIM5 mutants showed a normal accumulation in PML NBs (*Figure 3—figure supplement 1D*). Interestingly, the HIRA-HA mSIM2 showed a significant decrease in its accumulation in PML NBs with (*Figure 3F*) or without (*Figure 3—figure supplement 1C*) IFNβ treatment. This data confirms the importance of the SUMO-SIM interaction pathway in general, and at least, the putative SIM2 motif for the recruitment of the overexpressed ectopic HIRA in PML NBs regardless of the presence of IFNβ. Unfortunately, the levels of HIRA-HA mSIM2 expression remained very low at the cell population level compared to HIRA-HA WT (*Figure 3—figure supplement 1E*), preventing any biochemical analyses. Since SP100 is increased upon IFNβ treatment (*Figure 1F*) and is required for HIRA localization in PML NBs (*McFarlane et al., 2019*), we sought to refine further its contribution for HIRA accumulation in PML NBs. Knock-down of SP100 abrogated HIRA partitioning in PML NBs (*Figure 3—figure supplement 2A–B*), confirming previous data (*McFarlane et al., 2019*), without impacting DAXX localization at PML NBs (*Figure 3—figure supplement 2C*). Overexpression of an ectopic EYFP-SP100 protein rescued HIRA accumulation in PML NBs, as well as an ectopic EYFP-SP100 mutated on its main SUMOylation site K297R (*Figure 3—figure supplement 2D–F*). Overall, our data argue for a multistep molecular mechanism, involving PML SUMOylation, a putative SIM motif on HIRA, as well as SP100 regardless of its SUMOylation on K297R, in the accumulation of HIRA in PML NBs following activation of the IFN-I signaling pathway.

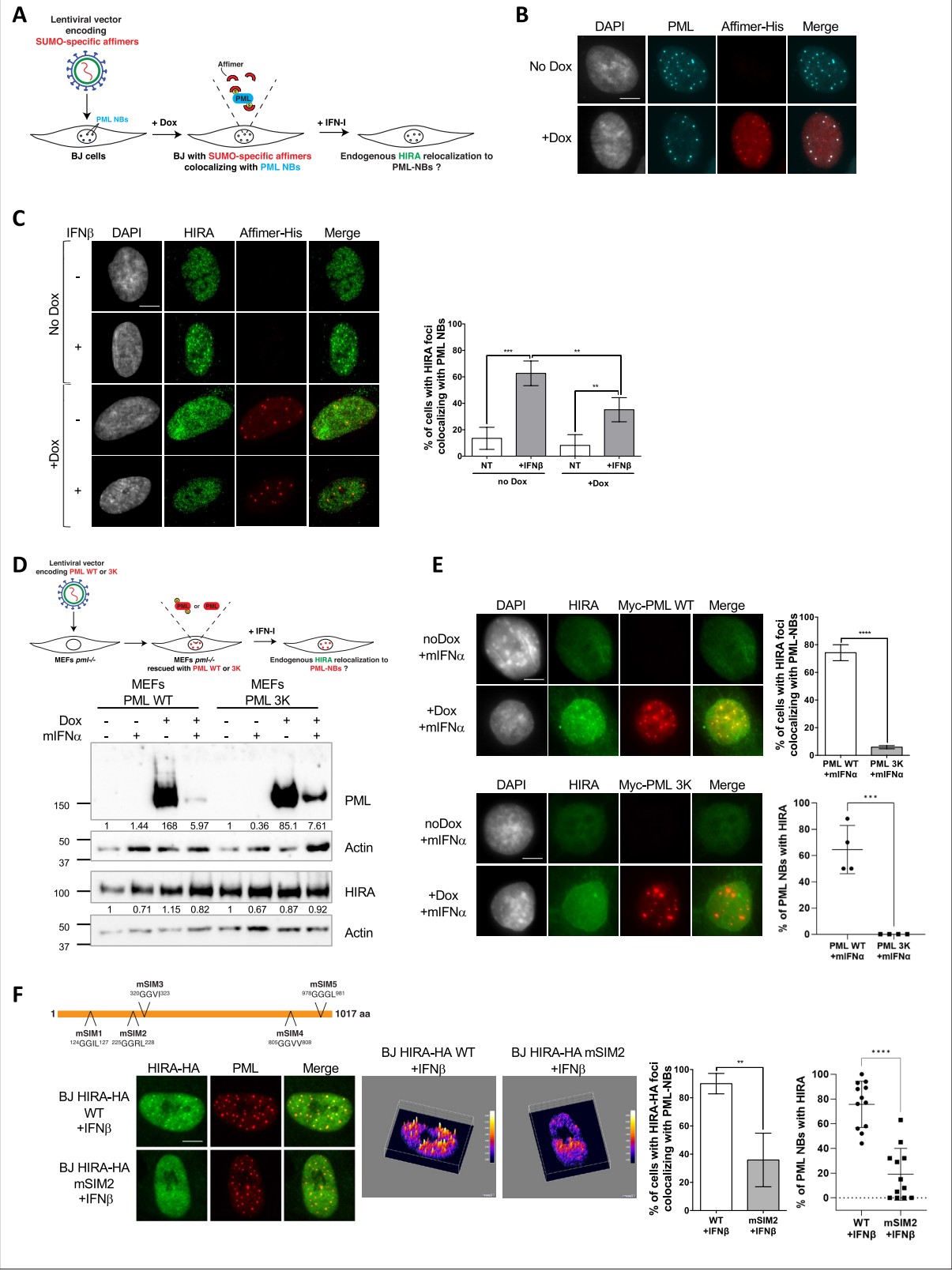

**Figure 3.** HIRA recruitment to PML NBs relies on SIM-SUMO interactions. (**A**) Experimental design to assess SUMO-specific Affimers impact on HIRA relocalization to PML NBs. BJ cells were transduced with a Dox-inducible lentiviral vector encoding for a 6xHis-tagged SUMO-specific S1S2D5 Affimer. When expressed, S1S2D5-His Affimers localize at PML NBs through their interactions with SUMOylated PML. (**B**) Fluorescence microscopy visualization PML (cyan) and S1S2D5-His Affimer marked with His antibody (red) in transduced BJ cells induced or not with doxycycline at 100 ng/mL for 30 hr.

*Figure 3 continued on next page*

*Figure 3 continued*

Colocalization of the S1S2D5-His Affimer (red) and PML NBs (cyan) produces white spots. Cell nuclei are visualized by DAPI staining (grey). Scale bar represents 10 µm. (**C**) (left) Fluorescence microscopy visualization HIRA (green) and S1S2D5-His Affimer marked with His antibody (red) in transduced BJ cells induced or not with doxycycline at 100 ng/mL for 30 hr and treated with IFNβ at 1000 U/mL for the last 24 hr. Cell nuclei are visualized by DAPI staining (grey). Scale bar represents 10 µm. (right) Histogram shows quantitative analysis of cells with HIRA localization at PML NBs. Numbers represent the mean of four independent experiments (± SD). p-values (Student t-test): **<0.01; ***<0.001. (**D**) (top) Experimental design to assess SUMOylated PML requirement for HIRA accumulation to PML NBs. MEFs *Pml*⁻/⁻ cells were transduced with Dox-inducible lentiviral vectors encoding for Myc-tagged WT or 3K non-SUMOylable PML proteins. Cells were then treated with murine type I IFNα and HIRA localization was observed by fluorescence microscopy. (bottom) Myc-PML proteins expression was verified by western blot analysis of human PML from total cellular extracts of MEFs cells describe above. HIRA proteins level was also verified. Actin is a loading control. Quantification of PML and HIRA levels relative to Actin are shown below the WB (numbers are representative of three independent experiments). (**E**) (left) Fluorescence microscopy visualization of HIRA (green) and Myc-PML with Myc antibody (red) on MEFs *Pml*⁻/⁻ cells rescued with Myc-tagged WT (top) or 3K (bottom) PML proteins through doxycycline treatment at 100 ng/mL for 24 hr. Cells were at the same time treated with murine IFNα at 1000 U/mL. Cell nuclei are visualized by DAPI staining (grey). Scale bars represents 10 µm. (right top) Histogram shows quantitative analysis of cells with HIRA localization at ectopic WT or 3K PML NBs in MEFs PML⁻/⁻ cells treated as on the left panel. Numbers represent the mean of three independent experiments (± SD). (right bottom) Histogram shows quantitative analysis of the percentage of PML NBs per cell showing colocalization with HIRA in MEFs PML⁻/⁻ cells treated as on the left panel. p-value (Student t-test): ***<0.001; ****<0.0001. (**F**) (top) Schematic representation of the localization of the mutations on putative SIM motifs on HIRA protein. (bottom left) Fluorescence microscopy visualization of HIRA-HA marked by HA antibody (green) and PML (red) in BJ cells stably transduced with HIRA-HA WT or HIRA-HA mSIM2 mutant and treated with IFNβ at 1000 U/mL for 24 hr. Scale bar represents 10 µm. (bottom middle) Graphics show HA signal intensity of each pixel within the nuclei depicted on the left panel in a 3D-surface plot. Higher expression signal appears in yellow to white colors. (bottom right) Histograms show quantitative analysis of cells with HIRA-HA localization in PML NBs (mean of three independent experiments [± SD]) or quantitative analysis of the percentage of PML NBs per cell showing colocalization with HIRA in BJ cells treated as on the bottom left panel. p-value (Student t-test): **<0.01; ****<0.0001.

The online version of this article includes the following source data and figure supplement(s) for figure 3:

**Source data 1.** Raw WB for *Figure 3D* for PML and Actin.

**Source data 2.** Raw WB for *Figure 3D* for PML and Actin with labels.

**Source data 3.** Raw WB for *Figure 3D* for HIRA.

**Source data 4.** Raw WB for *Figure 3D* for HIRA with labels.

**Source data 5.** Raw WB for *Figure 3D* for Actin.

**Source data 6.** Raw WB for *Figure 3D* for Actin with labels.

**Figure supplement 1.** HIRA accumulation in PML NBs depends on PML and on a putative SIM motif on HIRA.

**Figure supplement 1—source data 1.** Raw WB for *Figure 3—figure supplement 1A* for HIRA.

**Figure supplement 1—source data 2.** Raw WB for *Figure 3—figure supplement 1A* for HIRA with labels.

**Figure supplement 1—source data 3.** Raw WB for *Figure 3—figure supplement 1A* for PML.

**Figure supplement 1—source data 4.** Raw WB for *Figure 3—figure supplement 1A* for PML with labels.

**Figure supplement 1—source data 5.** Raw WB for *Figure 3—figure supplement 1A* for tubulin.

**Figure supplement 1—source data 6.** Raw WB for *Figure 3—figure supplement 1A* for tubulin with labels.

**Figure supplement 1—source data 7.** Raw WB for *Figure 3—figure supplement 1E* for HA.

**Figure supplement 1—source data 8.** Raw WB for *Figure 3—figure supplement 1E* for HA with labels.

**Figure supplement 1—source data 9.** Raw WB for *Figure 3—figure supplement 1E* for tubulin.

**Figure supplement 1—source data 10.** Raw WB for *Figure 3—figure supplement 1E* for tubulin with labels.

**Figure supplement 2.** HIRA accumulation in PML NBs depends on SP100, independent of its SUMOylation on K297R.

**Figure supplement 2—source data 1.** Raw WB for *Figure 3—figure supplement 2B* (left panel) for SP100 and tubulin.

**Figure supplement 2—source data 2.** Raw WB for *Figure 3—figure supplement 2B* (left panel) for SP100 and tubulin with labels.

**Figure supplement 2—source data 3.** Raw WB for *Figure 3—figure supplement 2B* (right panel) for HIRA.

**Figure supplement 2—source data 4.** Raw WB for *Figure 3—figure supplement 2B* (right panel) for HIRA with labels.

**Figure supplement 2—source data 5.** Raw WB for *Figure 3—figure supplement 2B* (right panel) for H3.3 and tubulin.

**Figure supplement 2—source data 6.** Raw WB for *Figure 3—figure supplement 2B* (right panel) for H3.3 and tubulin with labels.

**Figure supplement 2—source data 7.** Raw WB for *Figure 3—figure supplement 2F* (left panel) for SP100.

**Figure supplement 2—source data 8.** Raw WB for *Figure 3—figure supplement 2F* (left panel) for SP100 with labels.

**Figure supplement 2—source data 9.** Raw WB for *Figure 3—figure supplement 2F* (left panel) for tubulin.

**Figure supplement 2—source data 10.** Raw WB for *Figure 3—figure supplement 2F* (left panel) for tubulin with labels.

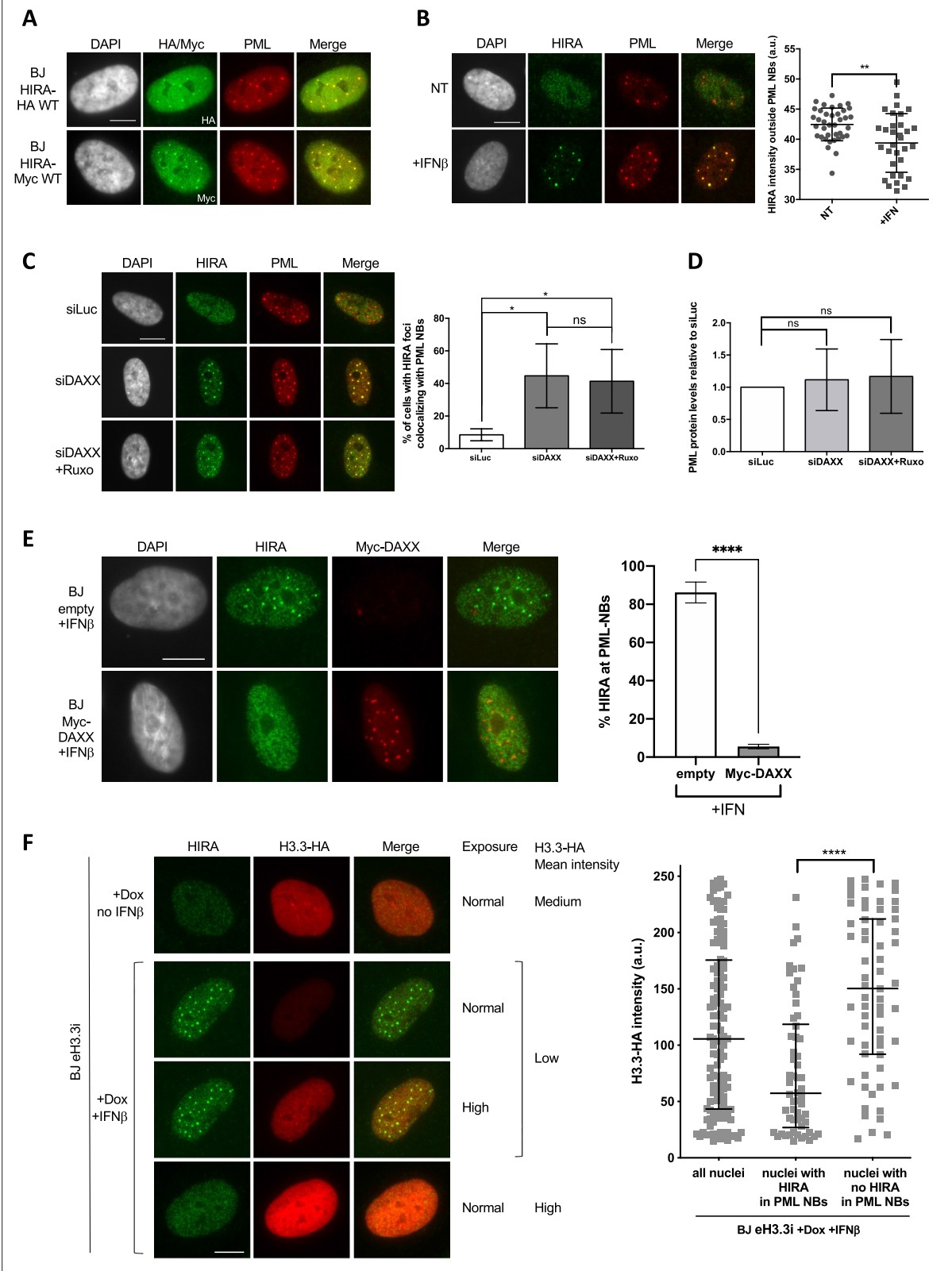

**Figure 4.** HIRA accumulation in PML NBs depends on DAXX and H3.3 levels. (**A**) Fluorescence microscopy visualization of HIRA-HA or HIRA-Myc marked respectively with HA and Myc antibodies (green) and PML (red) in BJ HIRA-HA WT and BJ HIRA-Myc WT cells. Cell nuclei are visualized by DAPI staining (grey). Scale bar represents 10 µm. (**B**) (left) Fluorescence microscopy visualization of HIRA (green) and PML (red) in BJ cells treated with IFNβ at 1000 U/mL for 24 hr (+IFNβ) or left untreated (NT). Cell nuclei are visualized by DAPI staining (grey). Scale bar represents 10 µm. (right) Histogram

*Figure 4 continued on next page*

*Figure 4 continued*

shows quantification of HIRA mean nuclear fluorescence intensity outside PML NBs (a.u: arbitrary units) in nuclei from five independent experiments. Bars represent median with interquartile range. p-values (Mann-Whitney u-test): **<0.01 (**C**) (left) Fluorescence microscopy visualization of HIRA (green) and PML (red) in BJ cells treated with the indicated siRNAs for 48 hr. Ruxolitinib (Ruxo) was added at 2 µM in the last 24 hr. Cell nuclei are visualized by DAPI staining (grey). Scale bars represent 10 µm. (right) Histogram shows quantitative analysis of cells with HIRA localization at PML NBs. p-values (Student t-test): *<0,05; ns: non significant. Numbers represent the mean of three independent experiments (± SD). (**D**) Histogram shows quantitative analysis of PML protein levels from western blot analysis presented in *Figure 4—figure supplement 1A*. p-values (Student t-test): ns: non significant. Numbers represent the mean of three independent experiments (± SD). (**E**) (left) Fluorescence microscopy visualization of HIRA (green) and Myc (red) in BJ cells transduced with a control retrovirus (empty) or with a retrovirus expressing Myc-DAXX for 48 hr. Cells were treated with IFNβ at 1000 U/ml in the last 24 hr. Cell nuclei are visualized by DAPI staining (grey). Scale bar represent 10 µm. (right) Histogram shows quantitative analysis of cells with HIRA localization at PML NBs. Numbers represent the mean of three independent experiments (± SD). p-value (Student t-test): ****<0.0001; ns: non significant. (**F**) (left) Fluorescence microscopy visualization of HIRA (green) and H3.3-HA marked by HA antibody (red) in BJ eH3.3i cells treated with doxycyclin at 100 ng/mL and with or without IFNβ at 1000 U/mL for 24 hr. High exposure indicates a lane where H3.3-HA signal was specifically increased in order to show H3.3-HA localization in PML NBs without saturating the signal in cells with higher expression. Scale bar represents 10 µm. (right) Quantification of nuclear H3.3-HA intensity levels in BJ eH3.3i cells treated as on the left panel. Mean H3.3-HA intensity levels were calculated on a pool of n=121 nuclei from three independent experiments. Nuclei were then separated on basis of accumulation of HIRA in PML NBs (nuclei with HIRA in PML NBs, n=58) or without it (nuclei with no HIRA in PML NBs, n=63) and mean H3.3-HA intensity was plotted for each category. Bars represent median with interquartile range. p-values (Mann-Whitney u-test): ****<0.0001.

The online version of this article includes the following source data and figure supplement(s) for figure 4:

**Figure supplement 1.** HIRA accumulation in PML NBs upon IFN-I treatment can be modulated by the pool of DAXX in primary cells.

**Figure supplement 1—source data 1.** Raw WB for *Figure 4—figure supplement 1A* for PML.

**Figure supplement 1—source data 2.** Raw WB for *Figure 4—figure supplement 1A* for PML with labels.

**Figure supplement 1—source data 3.** Raw WB for *Figure 4—figure supplement 1A* for DAXX and tubulin.

**Figure supplement 1—source data 4.** Raw WB for *Figure 4—figure supplement 1A* for DAXX and tubulin with labels.

**Figure supplement 1—source data 5.** Raw WB for *Figure 4—figure supplement 1A* for HIRA.

**Figure supplement 1—source data 6.** Raw WB for *Figure 4—figure supplement 1A* for HIRA with labels.

**Figure supplement 1—source data 7.** Raw WB for *Figure 4—figure supplement 1B* for DAXX.

**Figure supplement 1—source data 8.** Raw WB for *Figure 4—figure supplement 1B* for DAXX with labels.

**Figure supplement 1—source data 9.** Raw WB for *Figure 4—figure supplement 1B* for SP100.

**Figure supplement 1—source data 10.** Raw WB for *Figure 4—figure supplement 1B* for SP100 with labels.

**Figure supplement 1—source data 11.** Raw WB for *Figure 4—figure supplement 1B* for tubulin.

**Figure supplement 1—source data 12.** Raw WB for *Figure 4—figure supplement 1B* for tubulin with labels.

**Figure supplement 1—source data 13.** Raw WB for *Figure 4—figure supplement 1E* for Myc.

**Figure supplement 1—source data 14.** Raw WB for *Figure 4—figure supplement 1E* for Myc with labels.

**Figure supplement 1—source data 15.** Raw WB for *Figure 4—figure supplement 1E* for H3.3 and tubulin.

**Figure supplement 1—source data 16.** Raw WB for *Figure 4—figure supplement 1E* for H3.3 and tubulin with labels.

**Figure supplement 2.** HIRA accumulation in PML NBs upon IFN-I can be modulated by the pool of H3.3 histones in primary cells.

**Figure supplement 2—source data 1.** Raw WB for *Figure 4—figure supplement 2B* for HA and tubulin.

**Figure supplement 2—source data 2.** Raw WB for *Figure 4—figure supplement 2B* for HA and tubulin with labels.

## PML NBs serve as buffering sites for HIRA

So far, our data have shown that the accumulation of HIRA in PML NBs is correlated to an increase in PML/SP100 proteins upon IFN-I treatment. Importantly, ectopic expression of HIRA led to its accumulation in PML NBs without the need of IFN-I stimulation (*Ye et al., 2007b*; *Figure 4A*), suggesting that PML NBs could buffer an excess of HIRA proteins. In addition, quantification of nucleoplasmic levels of HIRA outside PML NBs showed a significant decrease after IFNβ treatment (*Figure 4B*), suggesting that PML NBs could act as sequestration places for HIRA upon IFN-I, leading to its substantial depletion from the surrounding nuclear space. We thus focused our attention on understanding the molecular mechanisms of this putative buffering function. Remarkably, PML NBs have a strong connection with the H3.3 chromatin assembly pathway. Soluble newly synthesized H3.3-H4 dimers localize in PML NBs in a DAXX-dependent manner in human primary cells, before deposition onto chromatin (*Corpet et al., 2014*; *Delbarre et al., 2013*). We thus reasoned that a modulation of the nucleoplasmic pool

of DAXX or H3.3-H4 could impact on the HIRA accumulation in PML NBs by competing for binding sites within PML NBs or by modulating the chaperone activity of HIRA, respectively. Knockdown of DAXX led to a significant increase of the baseline amounts of cells showing HIRA accumulation in PML NBs in the absence of IFN-I treatment (*Figure 4C*). This HIRA behavior was not affected by ruxolitinib, excluding a contribution of the IFN-I signaling pathway (*Figure 4C*, *Figure 4—figure supplement 1A*), but was still dependent on SP100 (*Figure 4—figure supplement 1B–C*). No significant increase for PML protein was observed confirming that PML concentration changes is not an absolute prerequisite for the accumulation of HIRA in PML NBs (*Figure 4D* and *Figure 4—figure supplement 1A*). The reverse approach consisting in the overexpression of an ectopic Myc-DAXX protein completely abrogates HIRA accumulation in PML NBs upon IFN-I treatment (*Figure 4E* and *Figure 4—figure supplement 1D–E*). Thus modulation of DAXX levels regulates HIRA accumulation in PML NBs.

To study the impact of soluble H3.3 levels on HIRA behavior, we used a cell line expressing an inducible HA-tagged form of H3.3 (BJ eH3.3i). Treatment with doxycyclin triggered a strong, yet highly variable, expression of eH3.3 (*Figure 4F*, *Figure 4—figure supplement 2A, B*), consistent with the polyclonal nature of the cells. As expected, doxycyclin did not impact HIRA localization in PML NBs in absence of IFNβ, and IFNβ triggered a normal accumulation of HIRA in PML NBs in these cells (*Figure 4—figure supplement 2A*). Induced expression of eH3.3 did not significantly impact on HIRA accumulation in PML NBs upon IFNβ at a cell population level (*Figure 4—figure supplement 2A*). However, close examination of individual cells showed an impaired accumulation of HIRA in PML NBs in cells with a strong expression of eH3.3, unlike low eH3.3 expressing cells (*Figure 4F*). Quantification of the mean eH3.3 nuclear fluorescence intensity showed that it was low on average in nuclei with accumulation of HIRA in PML NBs. On the contrary, nuclei with absence of HIRA in PML NBs correspond to those with a significant shift towards higher eH3.3 intensities (*Figure 4F*). Similar results were obtained in human primary lung fibroblasts excluding a cell-type effect (*Figure 4—figure supplement 2C*). These data highlight a strong antagonism between the presence of a H3.3 nucleoplasmic pool above a threshold, and HIRA accumulation in PML NBs upon IFNβ treatment. Thus, while PML NBs do not seem to play a major role as SUMOylation hotspots for HIRA, they function as buffering/sequestration places for HIRA dependent on the physiological state of the cell.

## PML depletion but not HIRA impairs ISGs expression

In order to look whether PML NBs could also function as regulatory hubs together with HIRA in the inflammatory response, we first analysed if HIRA and PML participate in ISGs transcriptional regulation. IFN-I is responsible for the upregulation of hundreds of ISGs as part of the innate immune response participating in the inhibition of virus replication (*Shaw et al., 2017*). We analyzed the expression of a selected set of ISGs, *MX1*, *OAS1*, *ISG15*, or *ISG54* (*IFIT2*) in cells treated with IFNβ for 6 or 24 hr. mRNA levels of these ISGs increased strongly after IFNβ treatment, peaking at 6 hr of treatment and slightly decreasing at 24 hr post addition of IFNβ (*Figure 5—figure supplement 1A*). Depletion of HIRA had no significant impact on ISGs expression at 6 and 24 hr of IFNβ stimulation (*Figure 5—figure supplement 1B–C*), consistent with previous reports (*McFarlane et al., 2019*; *Rai et al., 2017*). Compared to ISGs transcription climax at 6 hr post IFNβ, the peak of accumulation of HIRA in PML NBs at 24 hr post addition of IFNβ (*Figure 5—figure supplement 1D*) suggests that the latter might not be directly involved in ISGs transcriptional upregulation. In contrast, PML depletion led to a significant reduction in ISGs expression both at 6 and 24 hr after IFNβ stimulation (*Figure 5—figure supplement 1C*). *MX1*, *OAS1*, *ISG15*, or *ISG54* mRNA levels were only 27%, 16%, 28%, or 35% of the levels observed in 6 hr IFNβ-treated cells, respectively. Thus, these results suggest an essential role of PML in the IFN-I-dependent transcriptional upregulation of ISGs, which is independent of HIRA.

## Interferon-stimulated gene loci are juxtaposed to PML NBs after IFN-I stimulation

PML NBs make direct physical contacts with surrounding chromatin regions and these associations may serve to modulate genome functions and gene expression (*Corpet et al., 2020*). In the context of the IFNγ inflammatory response, genes within the MHCII locus are located in proximity of PML NBs (*Gialitakis et al., 2010*). Given the above results, we thought to examine the spatial connection between PML NBs and specific ISG loci. We performed immunostaining of the PML protein, together with fluorescence in situ hybridization (immuno-FISH) to detect the *PML*, *MX1*, *OAS1* gene loci in cells

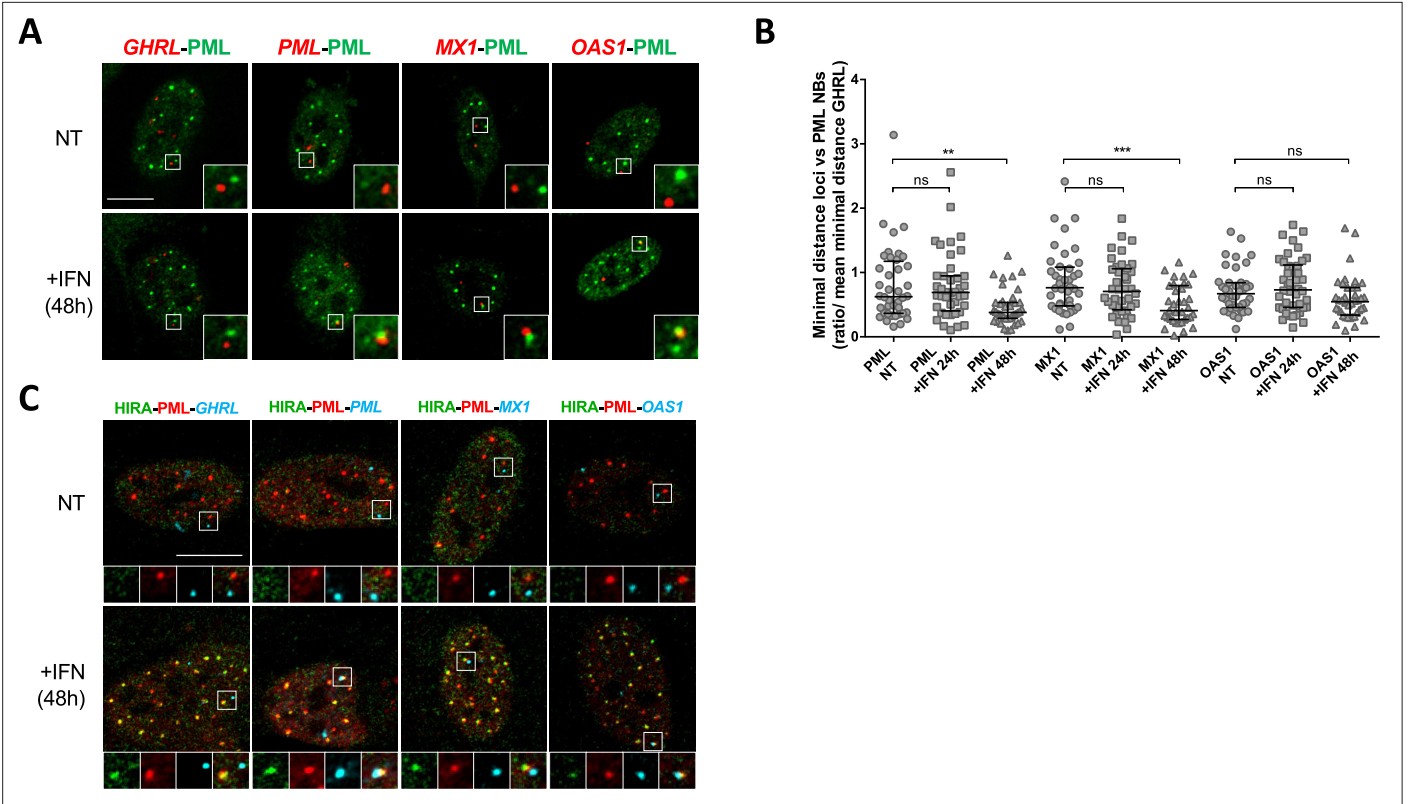

**Figure 5.** PML NBs become juxtaposed to ISGs loci upon IFN-I treatment. (**A**) Confocal fluorescence microscopy visualization of IF-FISH against PML proteins (green) and *GHRL* control gene locus (red) or *PML*, *MX1* or *OAS1* ISGs loci (red) in BJ cells treated with IFNβ at 1000 U/mL for 48 hr. Insets represent enlarged images (3 X) of selected areas and show the relative distance between one PML NB and one gene locus. Scale bar represents 10 μm. (**B**) Scatter plot shows the ratio of the minimal distance between PML NBs and ISGs loci on the mean minimal distance between PML NBs and *GHRL* control gene locus in nuclei from BJ cells treated or not with IFNβ at 1000 U/mL for the indicated time. The line in the middle represents the median of all observations. Results are from one representative experiment out of two experiments and are calculated on an average of 40 nuclei/condition. p-value (Mann-Whitney u-test): **<0.01; ***<0.001; ns: non significant. (**C**) Confocal fluorescence microscopy visualization of IF-FISH against HIRA (green) and PML proteins (red) and *GHRL* control gene locus (cyan) or *PML*, *MX1*, or *OAS1* ISGs loci (cyan) in BJ cells treated as in B. Insets and scale bar are as in B.

The online version of this article includes the following source data and figure supplement(s) for figure 5:

**Figure supplement 1.** ISGs are upregulated upon IFN-I treatment in a PML-dependent manner and become juxtaposed to PML NBs.

**Figure supplement 1—source data 1.** Raw WB for *Figure 5—figure supplement 1B* for HIRA.

**Figure supplement 1—source data 2.** Raw WB for *Figure 5—figure supplement 1B* for HIRA with labels.

**Figure supplement 1—source data 3.** Raw WB for *Figure 5—figure supplement 1B* for PML.

**Figure supplement 1—source data 4.** Raw WB for *Figure 5—figure supplement 1B* for PML with labels.

**Figure supplement 1—source data 5.** Raw WB for *Figure 5—figure supplement 1B* for tubulin.

**Figure supplement 1—source data 6.** Raw WB for *Figure 5—figure supplement 1B* for tubulin with labels.

treated or not with IFNβ. PML was used as a positive ISG control since previous immuno-trap analyses found a specific interaction between PML NBs and the *PML* gene locus upon IFNα treatment (*Ching et al., 2013*). To evaluate the specificity of potential spatial changes, we also scored localization of the Grehlin and Obestatin Prepropeptide (*GHRL*) locus, which is not an ISG (*Eggenberger et al., 2019*) and is localized in heterochromatin regions (*Becker et al., 2017*). Visual inspection showed an overall closer locus-to-PML NB proximity of *PML*, *MX1* and *OAS1*, but not *GHRL*, in IFNβ-treated cells relative to untreated cells (*Figure 5A*). To quantify the association of PML NB with ISG loci, we calculated the mean minimal distance (mmd) between each locus and the center of the closest PML NB per nucleus in untreated and treated cells. A decreased distance could be a consequence of the increased number and size of PML NBs upon IFNβ treatment (*Figure 5—figure supplement 1E*). We thus normalized the mmd for the ISGs to the one calculated for the *GHRL* locus. A marked decrease

in the mmd of PML NBs with the three loci was scored at 48 hr, which was significant for *PML* and *MX1* loci, reaching a calculated mmd of 0.67 μm and 0.72 μm, respectively, as compared to 1.06 μm for the *GHRL* locus (***Figure 5B***). We also confirm the presence of HIRA in PML NBs juxtaposed to ISGs by triple labelling (***Figure 5C***). The requirement of PML for the acute peak of transcription of ISGs at 6 hr of IFNβ (***Figure 5—figure supplement 1C***), in comparison to the occurrence of the juxtaposition of PML NBs with the ISGs loci at 48 hr post addition of IFNβ suggests that existing PML NBs are not directly involved in the transcriptional control of ISGs, but rather nucleate at ISGs loci from the PML proteins initially involved in ISGs transcription.

## IFN-I stimulation triggers accumulation of endogenous H3.3 in the 3' end region of transcribed ISGs

Using a tagged version of H3.3 in MEF cells, previous studies showed an increased and prolonged deposition of ectopic H3.3 in the transcription end sites (TES) region of ISGs upon IFN-I stimulation (***Sarai et al., 2013***; ***Tamura et al., 2009***). We thus wondered whether PML and HIRA could functionally impact endogenous H3.3 deposition on ISGs using an H3.3-specific antibody previously validated in ChIP (***Lee et al., 2019***). The amount of H3.3 remained unaffected by 24 hr of IFNβ stimulation excluding a putative ISG-like behavior (***Figure 6—figure supplement 1A***). We first investigated H3.3 incorporation on *MX1*, *OAS1*, and *ISG54* (*IFIT2*) by qPCR. Three distinct regions of the selected ISGs were analyzed: the promoter region, located just upstream (−120 pb) of the transcriptional start site (TSS), the middle of the coding region (mid), and a distal site in the coding region near the TES (see map in ***Figure 6A***). A slight decrease of H3.3 occupancy at promoter regions was measured (***Figure 6A***). The reduction following IFN-I stimulation likely reflects transcription-induced nucleosome depletion known to happen for many genes upon stimulation (***Workman, 2006***). Remarkably, IFNβ stimulation induced H3.3 incorporation most noticeably over the distal sites of the coding regions (***Figure 6A***). This was concomitant with an increase in H3K36me3, a histone mark added by the methyltransferase SETD2, which moves with RNA pol II during transcription (***Figure 6—figure supplement 1B***). Use of a control IgG antibody did not lead to any significant amount of immunoprecipitated DNA (% input) in any of the conditions highlighting the specificity of our ChIP experiment (***Figure 6—figure supplement 1C***). In addition, no change in H3.3 occupancy was observed at an enhancer region known to be enriched with H3.3 (***Pchelintsev et al., 2013***), underscoring the specificity of H3.3 accumulation in ISGs (***Figure 6A***). Normalization of H3.3 signal over the total H3 histones signal, which showed no major changes in histone density, confirmed the increased amount of H3.3 at ISGs with a preference for the TES regions (***Figure 6—figure supplement 1D***). This fits with the known replication-independent replacement of canonical H3 histones with H3.3 during transcription (***Ahmad and Henikoff, 2002***; ***Mito et al., 2005***; ***Workman, 2006***). No noticeable H3.3 increase was observed at representative mid or TES regions at 6 or 12 hr of IFNβ treatment (***Figure 6—figure supplement 1E***). Therefore, H3.3 increased deposition most likely takes place after the peak of ISGs transcription at 6 hr of IFNβ (***Figure 5—figure supplement 1A***). Importantly, H3.3 deposition continued to increase for an extended period of time and was even higher at 48 hr of IFNβ, suggesting that it could leave a long-lasting chromatin mark on ISGs (***Figure 6—figure supplement 1E***).

We then performed ChIP-Seq analysis for endogenous H3.3 on cells treated or not with IFNβ for 24 hr. We first examined H3.3 enrichment over the gene bodies of a published panel of equivalent sized ISGs (n=48) or non-ISGs (n=48) (***McFarlane et al., 2019***). The levels of H3.3 significantly increased on ISGs in IFNβ-treated cells with a clear bias towards the TES regions of the genes (***Figure 6B***). In contrast, no significant difference of H3.3 enrichment could be observed across the non-ISGs (***Figure 6C***). We selected genes with the highest difference in H3.3 enrichment at the TES region between IFNβ-treated and non-treated cells, and performed Gene Ontology (GO) analysis. GO analysis showed a clear enrichment in genes involved in IFNα and IFNγ response comforting the specific enrichment of H3.3 on the TES region of ISGs as a prolonged response to an IFN-I stimulus (***Figure 6D***). To evaluate the identity of H3.3-enriched genes in an unbiased manner, we performed an independent GO analysis on all genes found in the ChIP peak calls. This yielded similar results with IFNα and IFNγ responses being the most highly significant GO terms (***Figure 6—figure supplement 1F***). Thus, these findings establish that IFN-I triggers a specific long-lasting H3.3 deposition on ISGs following IFN-I stimulus.

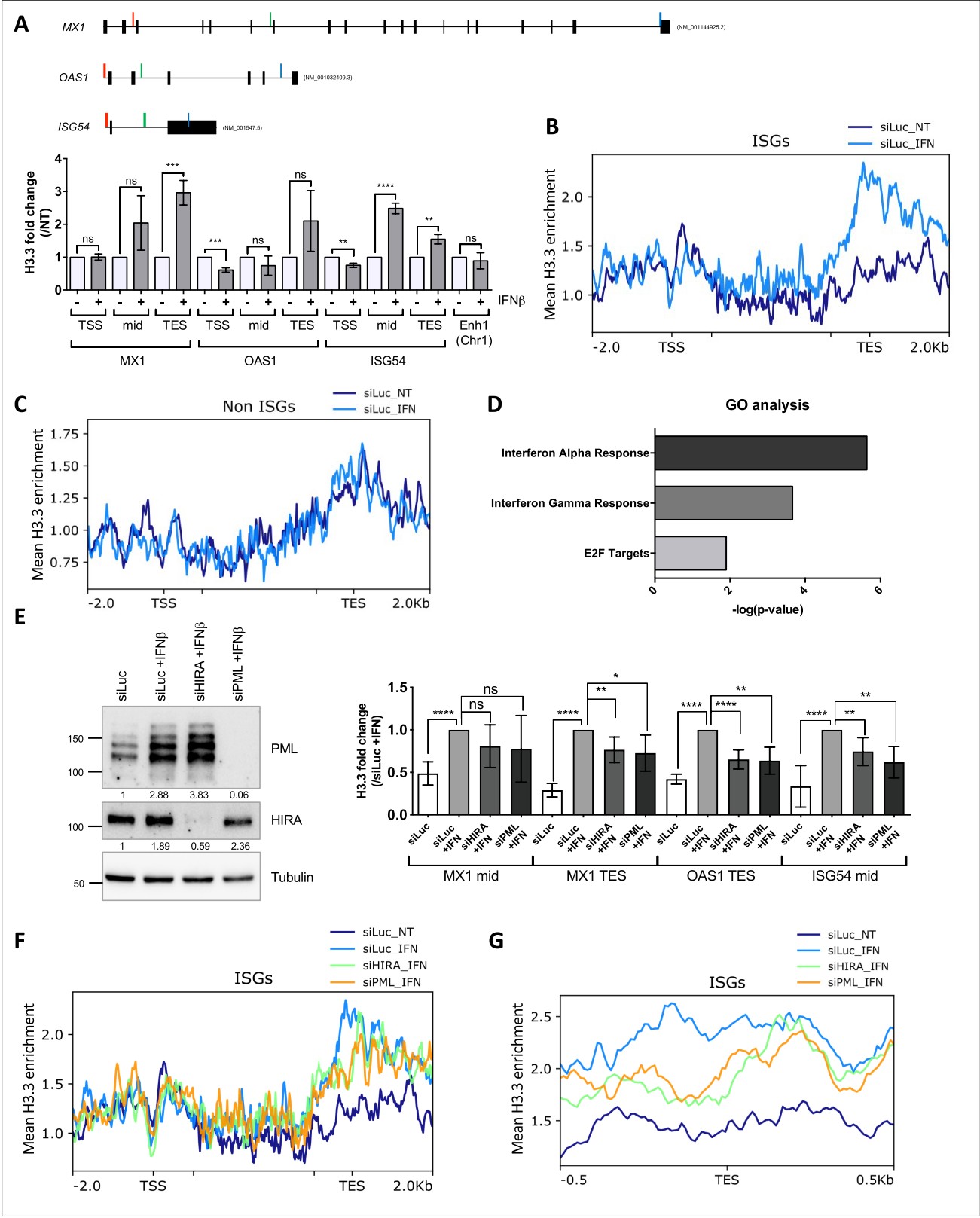

**Figure 6.** HIRA and PML depletions impair H3.3 enrichment at distal regions of ISGs. (**A**) (top) Schematic representation of *MX1*, *OAS1*, and *ISG54* gene loci. Localization of primers is marked in color: red, green, and blue for primers localized in the Transcription Start Site (TSS), mid or Transcription End Site (TES) region respectively. Black boxes represent exons and lines represent introns. (bottom) Histogram shows H3.3 enrichment fold change obtained through ChIP experiments on BJ cells treated or not with IFNβ at 1000 U/mL for 24 hr. Rationalization was performed on H3.3 enrichment in

*Figure 6 continued on next page*

*Figure 6 continued*

untreated cells. qPCR was performed on *MX1*, *OAS1*, and *ISG54* ISGs TSS, mid and TES regions and on one enhancer region on chromosome 1 (Enh1). Numbers represent the mean of three independent experiments (± SD). p-value (Student t-test): **<0.01; ***<0.001; ****<0.0001; ns: non significant. (**B**) ChIP-Seq profile of H3.3 enrichment over 48 core ISGs (*McFarlane et al., 2019*) ranging from –2.0 kb to 2.0 kb downstream and upstream of the gene bodies in BJ cells with siLuc for 72 hr and treated or not with IFNβ at 1000 U/mL for the last 24 hr of siRNA treatment. (**C**) ChIP-Seq profile of H3.3 enrichment over 48 coding non-ISGs equal in size to core ISGs (*McFarlane et al., 2019*), ranging from –2.0 kb to 2.0 kb downstream and upstream of the gene bodies (regions from TSS to +1000 bp and from –1000 to TES being kept unscaled) in BJ cells treated as in B. (**D**) Gene Ontology analysis on genes showing the highest differential H3.3 enrichment (log2(Fold Change)>5) in the TES +/-0.5 kb region between IFNβ treated and not treated conditions. (**E**) (left) Western blot analysis of HIRA and PML from total cellular extracts of BJ cells treated with the indicated siRNAs for 72 hr and with IFNβ at 1000 U/mL for the last 24 hr of siRNAs treatment. Tubulin is a loading control. Quantification of PML and HIRA levels relative to Tubulin are shown below the WB (numbers are representative of three independent experiments). (right) Histogram shows H3.3 enrichment obtained through ChIP experiments on BJ cells treated as on the left panel. Rationalization was performed on H3.3 enrichment in siLuc +IFN treated cells. qPCR was performed on *MX1* mid and TES regions, *OAS1* TES region and *ISG54* mid region. Numbers represent the mean of four independent experiments (± SD). p-values (Student t-test): *<0.05; **<0.01; ****<0.0001; ns: non significant. (**F**) ChIP-Seq profile of H3.3 enrichment over 48 core ISGs (*McFarlane et al., 2019*) ranging from –2 kb before TSS to 2 kb downstream the TES in BJ cells treated as in E. (**G**) ChIP-Seq profile of H3.3 enrichment over 48 core ISGs (*McFarlane et al., 2019*) ranging from –0.5 kb to 0.5 kb downstream and upstream of the TES in BJ cells treated as in E.

The online version of this article includes the following source data and figure supplement(s) for figure 6:

**Source data 1.** Raw WB for *Figure 6E* for PML.

**Source data 2.** Raw WB for *Figure 6E* for PML with labels.

**Source data 3.** Raw WB for *Figure 6E* for HIRA.

**Source data 4.** Raw WB for *Figure 6E* for HIRA with labels.

**Source data 5.** Raw WB for *Figure 6E* for tubulin.

**Source data 6.** Raw WB for *Figure 6E* for tubulin with labels.

**Figure supplement 1.** H3.3 and H3K36m3 increase on ISGs upon IFN-I.

**Figure supplement 1—source data 1.** Raw WB for *Figure 6—figure supplement 1A* for H3.3 and tubulin.

**Figure supplement 1—source data 2.** Raw WB for *Figure 6—figure supplement 1A* for H3.3 and tubulin with labels.

**Figure supplement 2.** Overexpression of HIRA increases H3.3 deposition at ISGs TES region upon IFN-I.

**Figure supplement 2—source data 1.** Raw WB for *Figure 6—figure supplement 2B* for HIRA and tubulin.

**Figure supplement 2—source data 2.** Raw WB for *Figure 6—figure supplement 2B* for HIRA and tubulin with labels.

**Figure supplement 2—source data 3.** Raw WB for *Figure 6—figure supplement 2B* for HIRA (high exposure).

**Figure supplement 2—source data 4.** Raw WB for *Figure 6—figure supplement 2B* for HIRA (high exposure) with labels.

## H3.3 deposition on ISGs is impaired upon HIRA or PML depletion

We next wondered whether HIRA and/or PML was essential for H3.3 deposition at ISGs. Cells were depleted of HIRA or PML (*Figure 6E*, left) and treated with IFNβ for 24 hr, before performing ChIP on H3.3. Knock-down of HIRA or PML led to a modest but consistent decrease in H3.3 at mid or TES regions of selected ISGs, suggesting the implication of these two proteins for the long-lasting H3.3 deposition on ISGs (*Figure 6E*, right). ChIP-Seq analysis confirmed a mild, but still significant, decrease in the loading of H3.3 at the TES on the panel of ISGs (*P*-value = 4,76e-03 for HIRA knock-down (KD) or 1.262e-03 for PML KD, as assessed by a paired Student's t-test) (*Figure 6F–G*). Representative *MX1*, *STAT1*, and *GCH1* genes, confirmed the deficit in H3.3 loading at the TES region of ISGs in the absence of HIRA or PML (*Figure 6—figure supplement 1G*). Of note, overexpression of HIRA, which increases the nucleoplasmic pool of HIRA as well as its accumulation in PML NBs (*Figure 4A*), triggered an increase in H3.3 loading upon IFN-I, as compared to cells with endogenous levels of HIRA (*Figure 6—figure supplement 2A–B*). We thus conclude that HIRA and PML both contribute to the increased long-lasting H3.3 deposition at the TES region of ISGs following the transcriptional peak associated to IFN stimulus.

## Accumulation of HIRA in PML NBs is not necessary for transcription-coupled H3.3 deposition at ISGs

Given the role of PML NBs in HIRA buffering, and their function as chromosomal hubs regulating transcription of associated ISGs and H3.3 deposition on these genes, we then wanted to investigate if these two functions were interconnected. PML is known to be required for HIRA loading on ISGs

as already published (*McFarlane et al., 2019*), and as confirmed by reanalysis of McFarlane data (*Figure 7—figure supplement 1A*). While proximity of PML NBs with ISGs could be a means to target HIRA on these genes, we hypothesize that HIRA buffering in PML NBs and its role in H3.3 deposition at ISGs upon IFN-I stress might be two independent events. We thus analyzed if the accumulation of HIRA in PML NBs was a prerequisite for the increased deposition of H3.3 on ISGs. We depleted SP100 that strongly impairs HIRA recruitment in PML NBs (*Figure 3—figure supplement 2A–F* and *McFarlane et al., 2019*). SP100 depletion did not prevent H3.3 loading at the ISGs TES upon IFN-β stimulation, but on the contrary increased it (*Figure 7A–B*). Of note, SP100 depletion led to an important increase in ISGs transcription (*Figure 7—figure supplement 1B–C*). This is consistent with its role as a general transcriptional repressor (*Seeler et al., 1998*), and with the higher H3.3 loading on ISGs in these conditions. Simultaneous knock-down of SP100 and HIRA in IFN-I treated cells (*Figure 7B*) reduced H3.3 deposition at the level of the control siRNA (*Figure 7A*). Thus, this demonstrates that the increase in H3.3 loading on ISGs TES upon SP100 knockdown and following IFN-β stimulation is mediated by HIRA independently of its localization in PML NBs.

## Discussion

There have been considerable efforts in defining the multiple roles of PML and PML NBs in the recent years including in chromatin dynamics. After having dissected the molecular mechanisms responsible for HIRA accumulation in PML NBs upon IFN-I treatment, we investigated the functional interplay of the PML NBs-HIRA-H3.3 axis in inflammatory response. Our work has revealed two independent roles for PML/PML NBs in (1) acting as buffering centers to modulate HIRA complex nucleoplasmic availability upon inflammatory stress and (2) regulating the transcriptional status of ISGs and the HIRA-mediated incorporation of H3.3 at these loci.

### HIRA accumulation in PML NBs upon inflammatory stresses is dependent on functional SUMO-SIM interactions and on SP100

While senescence was the first stress shown to induce accumulation of HIRA complex in PML NBs (*Banumathy et al., 2009*; *Jiang et al., 2011*; *Rai et al., 2011*; *Zhang et al., 2005*), IFN-I signaling pathway was recently shown to be responsible for similar behavior of HIRA upon a viral infection (*Cohen et al., 2018*; *McFarlane et al., 2019*; *Rai et al., 2017*). Here, we extend and corroborate these findings to various inflammatory stresses, including TNFα, or a synthetic dsRNA (PolyI:C) that increase the amount of PML and SP100 proteins. We show that HIRA partitioning in PML NBs is mediated by SUMO-SIM interactions, that can be inhibited by saturating PML/PML NBs SUMO sites with specific Affimers or a SIM-containing client such as DAXX. We identified a putative SIM motif on HIRA sequence that participate in its recruitment in PML NBs. Interestingly, the VLRL SIM motif identified is followed by a Serine in the position 229 (S229). Phosphorylation adjacent to SIM motifs can lead to an increased affinity towards SUMO1 lysine residues (*Cappadocia et al., 2015*). Other post-translational modifications such as phosphorylation could thus be important in regulating HIRA partitioning by changing the affinity between HIRA and SUMOylated PML proteins/partners. Of note, glycogen synthase kinase 3β (GSK-3β) mediated-phosphorylation of HIRA on S697 was suggested to drive HIRA accumulation in PML NBs upon senescence entry (*Ye et al., 2007a*). Our data using *Pml⁻/⁻* MEFs reconstituted with PML WT or PML 3K highlight the importance of PML main SUMOylation sites in recruiting HIRA complex. In addition, SP100, a general transcriptional repressor and a resident protein of PML NBs, is also critical for HIRA accumulation in PML NBs, independently of its SUMOylation. Overall, our data demonstrate that SUMO-SIM interactions are essential but not sufficient for HIRA accumulation in PML NBs after IFN-I treatment. Hence, the use of SUMO-specific Affimers opens interesting avenues to interfere with client recruitments in PML NBs including HIRA.

### PML NBs as buffering centers to regulate nucleoplasmic HIRA levels

Our study unveils an important aspect of the PML NBs-HIRA interplay, with PML NBs acting as buffering centers to regulate the excess pool of nucleoplasmic HIRA. First, previous studies (*Ye et al., 2007b*) confirmed by the present one, show that overexpression of an ectopic HIRA is sufficient to induce its accumulation in PML NBs in untreated cells. Second, we show that HIRA intensity level in the nucleus, outside PML NBs, decreases upon IFN-I treatment, while HIRA protein amounts slightly increases. Third, overexpression of DAXX totally abrogates HIRA accumulation in PML NBs upon IFN-I

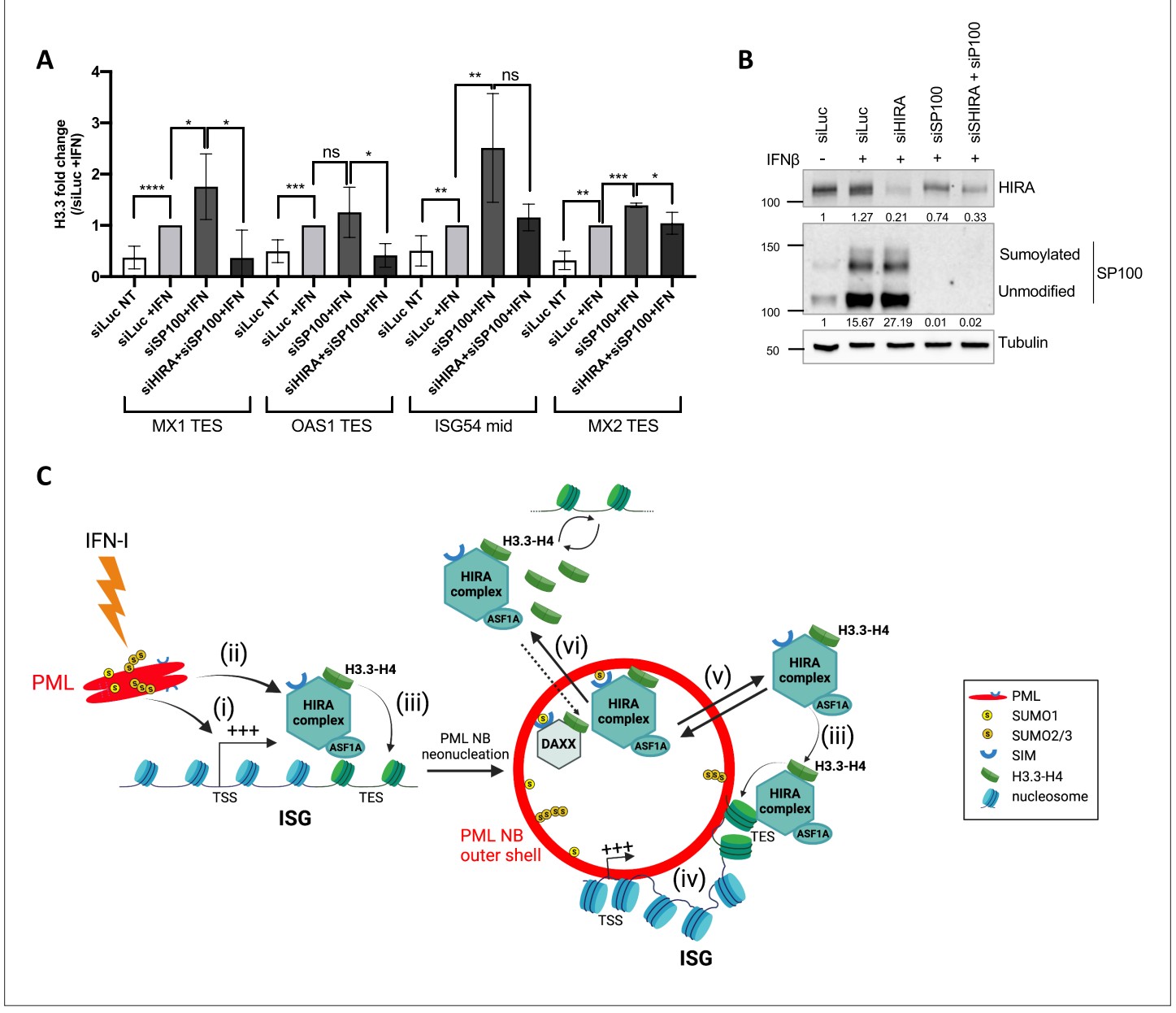

**Figure 7.** HIRA mediates H3.3 deposition during transcription of ISGs. (**A**) Histogram shows H3.3 enrichment obtained through ChIP experiments on BJ cells treated with the indicated siRNAs for 48 hr and with IFNβ at 1000 U/mL for the last 24 hr of siRNAs treatment. Rationalization was performed on H3.3 enrichment in siLuc +IFN treated cells. qPCR was performed on *MX1, OAS1,* and *MX2* TES regions as well as on *ISG54* mid region. Numbers represent the mean of three technical replicates out of two independent experiments (± SD). p-value (Student t-test): *<0.05; **<0.01; ***<0.001; ****<0.0001; ns: non significant. (**B**) Western blot analysis of HIRA and SP100 from total cellular extracts of BJ cells treated as in A. Tubulin is a loading control. Quantification of HIRA and SP100 levels relative to Tubulin are shown below the WB (numbers are representative of two independent experiments). (**C**) Model for the dual role of PML/PML NBs in inflammatory response. At early time points after an initial IFN-I stimulus, (i) PML is required for ISGs transcription and (ii) this could indirectly help to load HIRA on ISGs participating in H3.3 dynamics. (iii) While HIRA depletion does not affect ISGs transcription per se, it could participate in H3.3 deposition at ISGs, a function which does not seem to require its accumulation in PML NBs. (iv) PML neonucleation would mediate juxtaposition of PML NBs with ISGs at late times after IFN-I treatment which could help to keep a memory of the physiological state of the cell. (v) In addition, PML NBs play a second independent role by buffering the extra pool of HIRA complex available in the nucleus. (vi) Increase of DAXX protein levels could modulate the amount of available binding sites for HIRA within PML NBs or overexpression of the HIRA substrate H3.3 as a pool of free soluble H3.3 in the nucleoplasm could force HIRA out of PML NBs.

The online version of this article includes the following source data and figure supplement(s) for figure 7:

**Source data 1.** Raw WB for *Figure 7B* for HIRA.

**Source data 2.** Raw WB for *Figure 7B* for HIRA with labels.

*Figure 7 continued on next page*

*Figure 7 continued*

**Source data 3.** Raw WB for *Figure 7B* for SP100.

**Source data 4.** Raw WB for *Figure 7B* for SP100 with labels.

**Source data 5.** Raw WB for *Figure 7B* for tubulin.

**Source data 6.** Raw WB for *Figure 7B* for tubulin with labels.

**Figure supplement 1.** SP100 knock-down increases ISGs transcription upon IFN-I treatment.

treatment, suggesting that DAXX could occupy all available binding sites for HIRA partitioning in PML NBs. Fourth, DAXX knock-down induces a significant increase of untreated cells showing HIRA accumulation in PML NBs. These data support the hypothesis that the freeing of binding sites (in siDAXX treated cells) or the increase in binding sites (IFN-I treated cells) for HIRA might trigger its accumulation in PML NBs. Finally, overexpression of the HIRA substrate H3.3 as a pool of free soluble H3.3 in the nucleoplasm impairs HIRA accumulation in PML NBs upon IFN-I treatment.

The functionality of HIRA localization in PML NBs, first demonstrated during senescence (*Zhang et al., 2005*), remains to date elusive. An exciting hypothesis resulting from our data is that some acute stresses could massively retarget chromatin-bound HIRA from steady-state to specific stress-induced loci, leaving an extra pool of unbound HIRA, which then accumulates in PML NBs to be used for future duties. This buffering also depends on the amount of available binding sites for HIRA in PML NBs, as well as on the histone chaperone/cargo pool itself (HIRA/DAXX/H3.3 levels). Whether the sequestration of HIRA in PML NBs could as well impact its histone chaperone function elsewhere in the nucleus remains to be determined.

## PML regulates ISGs transcription and PML NBs associate with ISGs loci

A second important functional aspect unveiled in our study is the involvement of PML/PML NBs in the regulation of ISGs transcription, and the HIRA-mediated deposition of H3.3 at IFN-I stimulated ISGs. First, we show the importance of the PML protein for the initial burst of transcription of ISGs at 6 hr of IFNβ treatment. This is consistent with previous studies showing the association of PML NBs with nascent transcripts after IFNβ stimulation (*Fuchsová et al., 2002*), and the role of PML in ISGs induction following viral infection (*Alandijany et al., 2018*). PML proteins could be recruited to transcriptionally active ISGs by a specific, yet to be defined, protein-protein interaction. Previous studies showed that the nuclear DNA helicase II (NDH II), which is essential for gene activation, relocates in PML NBs in a transcription-dependent manner (*Fuchsová et al., 2002*). The authors suggested PML NBs could play a role in the transcriptional regulation of ISGs attached to PML NBs, although this was not investigated. Here, by using immuno-FISH, we demonstrate a juxtaposition of a subset of ISG loci with PML NBs at late time-points of IFNβ stimulation, confirming the potential role of PML NBs as regulatory chromosomal hubs. These data add to the likely importance of PML NBs-gene loci association for the regulation of specific sets of genes in a cell and stimulus-context manners, as exemplified recently for pluripotency-related genes in mESCs (*Sun et al., 2023*). Because PML targeting at specific gene loci is sufficient to induce de novo formation of PML NBs (*Brouwer et al., 2009*; *Chung et al., 2011*; *Erdel et al., 2020*; *Kaiser et al., 2008*; *Wang et al., 2018*), we hypothesize that chromatin-bound PML proteins involved in ISGs transcription could act as seeds to mediate neo-nucleation of PML NBs at ISG loci (see model in *Figure 7C*). Alternatively, displacement of ISGs-containing chromatin loops close to preexisting PML NBs could still be at play to explain this closer association upon IFNβ stimulation.

## H3.3-induced deposition at TES regions of transcribed ISGs is mediated by both HIRA and PML

Our data also reveal a role of the PML-HIRA axis in the H3.3 deposition on ISGs. Endogenous H3.3 deposition shows a strong preference for the ISGs TES regions, consistent with previous reports obtained in mouse cells overexpressing exogenous H3.3 (*Sarai et al., 2013*; *Tamura et al., 2009*). Also, our data highlight a long-lasting deposition of H3.3 up to 48 h after IFN-I stimulation, well beyond the peak of transcription of the ISGs. Deposition of endogenous H3.3 was reduced in the absence of PML consistent with the role of PML NBs in targeting H3.3 to chromatin (*Delbarre et al., 2013*), and in line with the role of PML in chromatinization of latent viral genomes (*Cohen et al., 2018*). Because

PML depletion impairs transcription of ISGs (*Figure 5—figure supplement 1C*; *Alandijany et al., 2018*), it could indirectly affect H3.3 deposition at TES regions, which is linked to the transcriptional activity of ISGs per se (*Sarai et al., 2013*). Thus, the implication of PML in the loading of HIRA on ISGs (*Figure 7—figure supplement 1A*; *McFarlane et al., 2019*) could be a result of the PML-regulated ISGs transcription, which could consequently impact on the HIRA-mediated H3.3 deposition on these loci. It is not unlikely that other H3.3 chaperones, such as the DAXX/ATRX complex, could compensate the absence of HIRA for H3.3 deposition as already observed for viral genomes chromatinization (*Cohen et al., 2018*). Alternatively, the remodeling protein CHD2, which has been shown to incorporate H3.3 on the promoters of myogenesis genes (*Harada et al., 2012*) could participate in H3.3 deposition at ISGs loci after IFN-I induction. Nonetheless, these data indicate the importance of a PML-HIRA axis to regulate H3.3 deposition on ISGs loci. Whether HIRA also mediates H3.3 recycling during transcription at ISGs via interaction with the RNA pol II (*Torné et al., 2020*; *Ray-Gallet et al., 2011*) and/or with an H3K36me3 methyltransferase (*Sarai et al., 2013*) should be investigated further. Thus, these data open interesting perspectives to study the maintenance of chromatin states at ISGs during inflammatory responses.

Finally, given the accumulation of HIRA in PML NBs upon IFNβ stimulation, an important question to address was whether this accumulation is required for HIRA-dependent H3.3 deposition at ISGs. The H3.3 ChIP analyses conducted in absence of HIRA accumulation in PML NBs upon IFN-I due to the depletion of SP100 showed an enrichment of H3.3 at ISGs loci, which decreased upon the additional knock-down of HIRA. These data support the importance of HIRA for the H3.3 deposition at ISGs, and favor the absence of correlation between HIRA accumulation in PML NBs and the deposition of H3.3 at ISGs loci, at least upon acute IFN-I stimulus. This further supports a role of PML NBs as nuclear depots for HIRA following an acute stress, possibly for its subsequent use after the acute stress resolution.

The role of the prolonged H3.3 deposition on ISGs can be multiple. First, this long-lasting mark could contribute to the acquisition of a functional IFN response memory. Indeed, H3.3 was shown to mediate memory of an active state upon nuclear transfer in *Xenopus laevis* (*Ng and Gurdon, 2008*). In addition, in MEFs, IFNβ stimulation creates a transcriptional memory of a subset of ISGs, which coincides with acquisition of H3.3 and H3K36me3 on chromatin (*Kamada et al., 2018*). A second stimulation with IFNβ allows a faster and greater transcription of so called "memory ISGs", which is dependent on H3.3 deposition during the first stimulation phase (*Kamada et al., 2018*). In HeLa cells, PML was shown to be required for the stronger re-expression of HLA-DRA after IFNγ restimulation, a locus that remained juxtaposed to PML NBs after transcription shut-off (*Gialitakis et al., 2010*). Second, H3.3 deposition may also serve to directly regulate ISGs expression. In mouse cells, H3.3 was found to be phosphorylated on Serine 31 on macrophages-induced genes following bacterial lipopolysaccharide stimulation, a post-translational mark serving as an ejection switch for the ZMYND11 transcriptional repressor, and allowing the transcriptional amplification of the target genes (*Armache et al., 2020*). Whether H3.3S31P is increased on ISGs upon IFN-I remains to be investigated.

Altogether, we propose a dual role for PML/PML NBs in regulating, on the one hand, HIRA nucleoplasmic pool and on the other hand, ISGs transcription and HIRA-mediated H3.3 deposition (see model in *Figure 7C*). PML NBs play a role of nuclear buffering centers for HIRA complex, to control its availability in the nucleoplasm, and thus possibly regulating its activity (*Figure 7C*). In addition, PML is required for ISGs transcription promoting H3.3 loading on these genes and could indirectly serve as a platform to load HIRA on ISGs (*Figure 7—figure supplement 1A*; *McFarlane et al., 2019*). While HIRA depletion does not affect ISGs transcription per se, it could participate in H3.3 deposition at ISGs, a function that is likely uncorrelated to its accumulation in PML NBs during an acute stress. Furthermore, as mentioned above, H3.3 deposition at ISGs after a first stimulus allows faster and greater transcription of ISGs upon restimulation (*Kamada et al., 2018*). Juxtaposition of PML NBs with ISGs at late times after IFN-I treatment could help to keep a memory of the physiological state of the cell. PML would remain in close proximity to ISGs to regulate them upon a second wave of IFN-I stimulation and HIRA accumulation in PML NBs could also be a mean for the cell to make the chaperone complex available much faster in case of a second inflammatory wave.

In conclusion, our study highlights two important functional and independent roles for PML NBs in the inflammatory response, which add to their pivotal involvement in various stress responses.

## Methods

### Cell lines and retro/lentiviruses production

Human BJ primary foreskin fibroblasts (ATCC, CRL-2522), human IMR90 fetal lung fibroblasts (ATCC, CCL-186), human HEK 293T embryonic kidney cells (Intercell, AG) and mouse MEFs embryonic fibroblasts $Pml^{+/+}$ or $Pml^{-/-}$ (from Dr. Lallemand-Breitenbach, and whose cell identity was authenticated by STR profiling) were cultivated in DMEM medium (Sigma-Aldrich, D6429) containing 10% of fetal calf serum (FCS) (Sigma-Aldrich, F7524), 1% of penicillin/streptomycin (Sigma-Aldrich, P4458), at 37 °C under 5% CO2 and humid atmosphere. All cell lines were tested negative for mycoplasma contamination. Drugs and molecules used for cell treatments are described in the **Key Resources Table** in Appendix (duration is mentioned in the main text). BJ, MEFs or IMR90 cells stably expressing transgenes were obtained by retroviral or lentiviral transduction as in *Cohen et al., 2018*. Transduced cells were then selected 24 h later by adding the appropriate selective drug (puromycin (Invivogen, ant-pr) at 1 µg/mL or blasticidin (Invivogen, ant-bl) at 5 µg/mL).

### Plasmids

Plasmids are described in the **Key Resources Table** in Appendix. Tat-S1S2D5-Flag-His Affimer (Tat: nuclear localization sequence), obtained by PCR using pcDNA5-Tat-S1S2D5-Flag-His as template (graciously sent by Dr. David J. Hughes *Hughes et al., 2017*), was cloned in puromycin resistant pLVX-TetOne plasmid.

HIRA WT, obtained by RT-PCR from HeLa cells, was cloned in blasticidin resistant pLentiN plasmid with addition of HA or Myc tag in the C-terminus. HIRA-HA mSIM and K809G mutants were obtained by site-directed mutagenesis using QuickChange Lightning Site-directed Mutagenesis kit (Agilent Technologies, #210518). SIM motifs are characterized by a group of hydrophobic amino acids ((V/I/L)x(V/I/L)(V/I/L) or (V/I/L)(V/I/L)x(V/I/L)). HIRA mSIM mutant sequences are the following: mSIM1: aa $^{124}$VSIL$^{127}$ mutated in $^{124}$GGIL$^{127}$, mSIM2: aa $^{225}$VLRL$^{228}$ mutated in $^{225}$GGRL$^{228}$, mSIM3: aa $^{320}$LLVI$^{323}$ mutated in $^{320}$GGVI$^{323}$, mSIM4: aa $^{805}$VVVV$^{808}$ mutated in $^{805}$GGVV$^{808}$, mSIM5: aa $^{978}$VVGL$^{981}$ mutated in $^{978}$GGGL$^{981}$.

Myc-PML1 WT and 3K mutant, obtained by PCR using pLNGY-PML1 and pLNGY-PML1.KKK as template (kind gift by Dr. Roger Everett), were cloned in puromycin resistant pLVX-TetOne plasmid. H3.3-SNAP-HA3 obtained by PCR using pBABE-H3.3-SNAP-HA3 as template (kind gift by Dr. Lars Jansen) was cloned into puromycin-resistant pLVX-TetOne plasmid with EcoRI restriction enzyme. Myc-DAXX cloned into pLNCX2 was described in *Corpet et al., 2014*. EYFP-SP100 isoform A WT and K297R mutant (containing a cDNA that is resistant to the siRNA against SP100), obtained by PCR using pLNGY-EYFP-SP100 WT and K297R (kind gift of Dr Roger Everett), were cloned into puromycin resistant pLVX-TetOne plasmid.

### siRNAs

BJ cells were transfected with 40–60 nM of human siRNA for different timings (indicated in the main text for each experiment) using Lipofectamine RNAiMax reagent (Invitrogen, 13778–075) and Opti-MEM medium (Gibco, 31-985-070). siRNAs used and their sequences are summarized in the **Key Resources Table** in Appendix. siSUMO1 and siSUMO-2/3, were co-transfected into BJ cells at 30 nM each. siSP100 targets all SP100 isoforms (*Everett et al., 2008*).

### Antibodies

All the primary antibodies used in this study, together with the species, the references and the dilutions for immunofluorescence and western blotting, are summarized in the **Key Resources Table** in Appendix.

### Immunofluorescence (IF)

Immunofluorescence was performed as in *Corpet et al., 2014* (see the **Key Resources Table** in Appendix for antibodies dilution). Highly cross-absorbed goat anti-mouse or anti-rabbit (H+L) Alexa-488, Alexa-555 or Alexa-647 (Invitrogen) were used as secondary antibodies. Cells were then incubated in DAPI (Invitrogen Life Technologies, D1306) diluted in PBS at 0.1 µg/mL for 5 min at RT°C.

Coverslips were mounted in Fluoromount-G (SouthernBiotech, 0100–01) and stored at 4 °C before observation.

## Proximity ligation assay (PLA)

Proximity Ligation Assays were performed with the Duolink In Situ Red Starter Kit Mouse/Rabbit (Sigma-Aldrich, DUO92101). Cells on coverslips were fixed in 2% PFA for 12 min at RT°C and then permeabilized in PBS 0.2% Triton X-100 for 5 min at RT°C. Cells were then treated according to the manufacturer's instructions (see the **Key Resources Table** in Appendix for dilutions of primary antibodies). Coverslips were mounted in Duolink In Situ Mounting Medium with DAPI and stored at 4 °C before observation.

## Immunofluorescence - Fluorescence in situ hybridization (IF-FISH)

FISH probes were generated from different BACs: RP11-438J1, RP11-185E17, RP11-120C17 and RP11-134B23 BAC clones for GHRL, PML, MX1 and OAS1, respectively. Briefly, 1 µg of BAC were incubated for nick-translation with 4.3 ng of DNAse I (Roche, 104159), 7 U of DNA polymerase (Promega, M2051), dithiothreitol (DTT) at 10 µM, dATP, dTTP and dGTP at 40 µM each (Thermo Scientific, R0141/R0161/R0171), dCTP at 10 µM (Thermo Scientific, R0151) and Cy3 labelled dCTP at 10 µM (Cytiva, PA53021). Nick-translation was performed for 3 hr at 15 °C and stopped by an incubation at 72 °C for 10 min. Size of generated probes were verified on agarose gel. Probes were then mixed with 20 µg of COT Human DNA (Roche, 11 581 074 001) and 79 µg of Salmon sperm DNA (Invitrogen, 15632–011). Volume was completed with TE buffer (10 mM Tris-HCl pH 8, 1 mM EDTA). DNA was precipitated with 300 mM of sodium acetate and 70% of chilled EtOH for 2 hr at –20 °C. DNA pellets were resuspended in formamide at 20 ng/µL final concentration.

After performing classic immunofluorescence as described above (without the DAPI staining step), cells were post-fixed in 2% PFA for 12 min at RT°C and then permeabilized and deproteinized in PBS 0.5% Triton X-100 0.1 M HCl for 10 min at RT°C. Samples were dehydrated in successive EtOH baths (2x70% EtOH, 2x85% EtOH and 2x100% EtOH). After co-denaturation at 80 °C for 5 min, cells' DNA was hybridized with FISH probes diluted at 1/5 O/N at 37 °C in dark and humid chamber. Cells were then washed 5 min in Saline-Sodium Citrate (SSC) 0.5 X at 68 °C, 2 min in SSC 1 X at RT°C and incubated in DAPI diluted in SSC 2 X for 5 min at RT°C. Coverslips were mounted in Fluoromount-G and stored at 4 °C before observation.

## Microscopy, imaging, and quantification

Images were acquired with the Axio Observer Z1 inverted wide-field epifluorescence microscope (100 X or 63 X objectives/N.A. 1.46 or 1.4) (Zeiss) and a CoolSnap HQ2 camera from Photometrics. Identical settings and contrast were applied for all images of the same experiment to allow data comparison. Raw images were treated with Fiji software or with Photoshop (Adobe). HIRA complex accumulation in PML NBs was attested by manual counting of a minimum of 100 cells for each condition and per replicate. The percentage of PML-NBs with HIRA localization in individual cells was measured using the CellProfiler Primary Objects Identification and Objects Relation functions. PML-NBs and genes loci proximity was measured using the Fiji RenyiEntropy mask on PML and FISH staining. X and Y coordinates for the center of the spots were recovered and all distances between each PML NBs and gene loci were calculated using the formula $d = \sqrt{(x1 - x2)^2 + (y1 - y2)^2}$ to find the minimal distance in each nucleus. Quantification of nuclear intensities was performed with Fiji. Briefly, DAPI and PML stainings were used to define masks of nuclei and of PML NBs. We quantified mean HA fluorescence intensity within each nucleus with the measure function applied on the red (HA) channel. To quantify HIRA intensity outside PML NBs, we first created a mask of nuclei devoid of PML NBs (Image calculator function of Fiji) and then applied the measure function on HIRA channel.

## Western blotting (WB)

Total cellular extracts were obtained by directly lysing the cells in 2 X Laemmli sample buffer (LSB) (125 mM Tris-HCl pH 6.8, 20% glycerol, 4% SDS, bromophenol blue) containing 100 mM DTT. RIPA extracts were obtained by lysing the cells in RIPA buffer (50 mM Tris-HCl pH 7.5, 150 mM NaCl, 0.5% Na-Deoxycholate, 1% NP-40, 0.1% SDS, 5 mM EDTA) supplemented with 1 X protease inhibitor

cocktail (PIC) for 20 min on ice. After incubation, RIPA extracts were centrifugated for 10 min at 16,000 *g* at 4 °C and supernatants were recovered and diluted with 4 X LSB.

Western Blot was performed as in *Corpet et al., 2014* (see the **Key Resources Table** in Appendix for antibodies dilution). Signal was revealed on ChemiDoc Imaging System (Bio-Rad) by using Amersham ECL Prime Western Blotting Detection Reagent (GE Healthcare Life Sciences, RPN2236) or Clarity Max Western ECL Blotting Substrate (Bio-Rad, 1705062).

## Chromatin immunoprecipitation (ChIP)

Cells were crosslinked directly in the culture dishes according to *Becker et al., 2017*. After the PBS washes, cell pellets were snap-frozen in liquid nitrogen and stored at –80 °C before immunoprecipitation. Cells were de-frozen on ice and chromatin was prepared following the TruChIP protocol from Covaris, as described in *Cohen et al., 2018*. We used the Covaris M220 Focused-ultrasonicator to shear through chromatin (7 min at 140 W, Duty off 10%, Burst cycles 200). After shearing, chromatin immunoprecipitation was performed as in *Becker et al., 2017*. We used 20 µL of protein A magnetic dynabeads (Invitrogen, 10001D) for immunoprecipitation with 2 µg of the following rabbit primary antibodies: anti-H3.3 (Diagenode, C15210011), anti-panH3 (Abcam, ab1791), rabbit IgG (Diagenode, C15410206). After DNA purification according to *Becker et al., 2017*, DNA pellets were resuspended in ddH20 and stored at –20 °C before qPCR analysis.

## Reverse transcription (RT)

TRIzol reagent protocol (Invitrogen, 15596026) was used to isolate total RNAs, resuspended in ddH2O according to the manufacturer instructions. Contaminant DNA was removed with the DNA-free DNA Removal Kit (Invitrogen, AM1906). We used 1 µg of RNA for reverse transcription (RT). RNAs were annealed with Random Primers (Promega, C118A) and RT was performed with the RevertAid H Minus Reverse Transcriptase (Thermo Scientific, EP0452) according to the manufacturer instructions. cDNAs were stored at –20 °C before qPCR analysis.

## Quantitative PCR (qPCR)

qPCRs were performed using the KAPA SYBR qPCR Master Mix (SYBR Green I dye chemistry) (KAPA BIOSYSTEMS, KK4618). Primers used for qPCR are described in the **Key Resources Table** in Appendix.

## ChIP-Seq analysis

After ChIP, libraries were made in BGI and sequenced on a BGISEQ-500 sequencing platform (https://www.bgi.com). An average of 34 Million single-end 50 bp reads was obtained for each library. Reads were trimmed using Trimmomatic and quality assessed with FastQC. Reads were aligned to the human genome hg38 using the BWA alignment software. Duplicate reads were identified using the picard tools script and only non-duplicate reads were retained. Broad peaks calling was performed with MACS2 (*Zhang et al., 2008*) (`"--extsize 250 -q 0.01 --broad --broad-cutoff 0.05"`), using input DNA as control. We defined all possible locations of H3.3 by merging broad peaks identified in our four conditions (n=190295), and annotated them with Homer (http://homer.ucsd.edu/homer/download.html). We counted reads extended to 250 bp falling into these possible locations, in the four ChIP and their corresponding inputs, using bedtools-intersect. CPMs were obtained by dividing raw counts by the total number of mapped reads normalized to 1e6, and RPKMs by dividing CPMs by the peak length normalized to 1e3. Input RPKMs, used as background, was substracted from the respective ChIP RPKMs. We focused on 0.5% of the peaks with highest RPKM difference (n=951) between IFNβ treated and not treated conditions, of which 711 were intragenic. These peaks allowed us to defined a set of 654 genes, on which we performed GO analysis, with MsigDB, using enrichR plaform (*Kuleshov et al., 2016*).

As a complementary approach, we measured the ChIP enrichment within the 1000 bp regions spanning the TESs (–500+500), extending all unique reads into 250 bp fragments, and counting those falling within TES using bedtools-intersect. CPMs were obtained similarly, and input DNA CPMs, used as background, was substracted from ChIP CPMs. Genes with the log2 of differential TES enrichment between IFNβ treated and not treated conditions being higher than 5 (log2(Fold Change)>5) were retained for GO analysis, as described above.

PlotProfile were generated using the DeepTools suite, starting from the MACS2 fold enrichment bigwig files, which take into account the read extension, the input DNA background and the library size normalization. The list of 48 core ISGs and 48 non-ISGs equal in size to the core ISGs was taken from *McFarlane et al., 2019*. In order to reduce the noise on the profiles, we selected for each gene the transcript with the highest H3.3 enrichment at the TES in the IFNβ treated condition. Genome browser snapshots of H3.3 enrichment were generated using Integrative Genomics viewer (IGV: https://software.broadinstitute.org/software/igv/).

We re-analysed the HIRA ChIP-Seq dataset from *McFarlane et al., 2019*. Rapidly, fastq files were downloaded from GEO databank under the accession number GSE128173 and re-aligned to the human genome 19 (hg19) using Bowtie2 (*Langmead and Salzberg, 2012*). MACS2 fold enrichment bigwig files were then used with the DeepTools suite (*Ramírez et al., 2016*) to create profile-plots of the 48 core ISG used in *McFarlane et al., 2019*.

### Statistical analyses and figures

Histograms and statistical analyses were performed using GraphPad Prism 6. To perform Student t test, we verified normal distribution of samples using Shapiro test and variance equality with Fisher test. Mann-Whitney u-test was applied in absence of normality for the sample distribution. p-Values are depicted on graphs as follows: $*<0.05$; $**<0.01$; $***<0.001$; $****<0.0001$. Biorender.com was used to generate figures schemes and model.

### Materials availability

All plasmids and cell lines generated in this study can be accessed upon request to the corresponding authors.

## Acknowledgements

We thank Dr. Valérie Lallemand-Breitenbach for the *Pml* WT and *Pml-/-* MEFs. We thank Dr. Chris Boutell for the pLVX-His-SUMO1/2/3 plasmids. We thank Dr. Roger Everett for the PML- and SP100-containing plasmids. We thank Dr. David J Hughes for the SUMO-specific Affimers plasmids. We thank Dr. Lars Jansen for the pBABE-H3.3-SNAP-HA3 plasmid. We thank Dr. Caroline Schluth-Bolard for her kind help with the FISH on ISGs and for the GHRL BAC.

PL laboratory is funded by grants from the Centre National de la Recherche Scientifique (CNRS), Institut National de la Santé et de la Recherche Médicale (INSERM), University Claude Bernard Lyon 1, French National Agency for Research-ANR [EPIPRO ANR-18-CE15-0014-01, CHROMACoV ANR-20-COV9-0004; IFN-Epi-IM ANR-21-CE17-0018]; LabEX DEVweCAN and DEV2CAN [ANR-10-LABX-61]; AFM-Téléthon Plans stratégiques MyoNeurALP & MyoNeurALP2, the Comité départemental du Rhône de La Ligue contre le Cancer and the Fondation pour la Recherche Médicale (FRM) (grant number FDT202001010820 to CK). PL is a CNRS Research Director and AC is assistant professor in the University Claude Bernard Lyon 1.

## Additional information

### Funding

| Funder | Grant reference number | Author |
| --- | --- | --- |
| Centre National de la Recherche Scientifique | | Patrick Lomonte |
| Institut National de la Santé et de la Recherche Médicale | | Patrick Lomonte |
| Université Claude Bernard Lyon 1 | | Patrick Lomonte |
| Agence Nationale de la Recherche | EPIPRO ANR-18-CE15-0014-01 | Patrick Lomonte |

| Funder | Grant reference number | Author |
|---|---|---|
| Agence Nationale de la Recherche | CHROMACoV ANR-20-COV9-0004 | Patrick Lomonte |
| Agence Nationale de la Recherche | IFN-Epi-IM ANR-21-CE17-0018 | Patrick Lomonte Armelle Corpet |
| LabEx DEvweCAN | | Patrick Lomonte |
| Agence Nationale de la Recherche | ANR-10-LABX-61 | Patrick Lomonte |
| AFM-Telethon | Plans stratégiques MyoNeurALP & MyoNeurALP2 | Patrick Lomonte |
| Comite departemental du Rhone de La Ligue contre le Cancer | | Armelle Corpet |
| Fondation pour la Recherche Médicale | FDT202001010820 | Constance Kleijwegt |

The funders had no role in study design, data collection and interpretation, or the decision to submit the work for publication.

## Author contributions

Constance Kleijwegt, Conceptualization, Formal analysis, Validation, Investigation, Visualization, Methodology, Writing – original draft, Writing – review and editing; Florent Bressac, Coline Seurre, Validation, Investigation; Wilhelm Bouchereau, Formal analysis; Camille Cohen, Conceptualization, Investigation; Pascale Texier, Investigation; Thomas Simonet, Data curation, Software, Formal analysis, Visualization; Laurent Schaeffer, Supervision; Patrick Lomonte, Conceptualization, Supervision, Funding acquisition, Validation, Methodology, Writing – original draft, Project administration, Writing – review and editing; Armelle Corpet, Conceptualization, Formal analysis, Supervision, Funding acquisition, Validation, Investigation, Visualization, Methodology, Writing – original draft, Writing – review and editing

## Author ORCIDs

Constance Kleijwegt http://orcid.org/0000-0002-9329-0806
Armelle Corpet http://orcid.org/0000-0002-2126-5783

## Decision letter and Author response

Decision letter https://doi.org/10.7554/eLife.80156.sa1
Author response https://doi.org/10.7554/eLife.80156.sa2

# Additional files

## Supplementary files

• MDAR checklist

## Data availability

The ChIP-Seq datasets have been deposited in the Gene Expression Omnibus (GEO; http://www.ncbi.nlm.nig.gov/geo/) under the accession number GSE233298.

The following previously published dataset was used:

| Author(s) | Year | Dataset title | Dataset URL | Database and Identifier |
|---|---|---|---|---|
| Kleijwegt C, Corpet A, Lomonte P, Simonet T | 2021 | Interplay between PML NBs and HIRA for H3.3 deposition on transcriptionally active interferon-stimulated genes | https://www.ncbi.nlm.nih.gov/geo/query/acc.cgi?acc=GSE183937 | NCBI Gene Expression Omnibus, GSE183937 |

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

# Appendix 1

## Appendix 1—key resources table

| Reagent type (species) or resource | Designation | Source or reference | Identifiers | Additional information |
|---|---|---|---|---|
| Cell line (*H. sapiens*) | BJ | ATCC | CRL-2522 | Human primary foreskin fibroblasts |
| Cell line (*H. sapiens*) | BJ S1S2D5 | This paper | | Human primary foreskin fibroblasts transduced with TetOne-Tat-S1S2D5-Flag-His |
| Cell line (*H. sapiens*) | BJ eH3.3i | This paper | | Human primary foreskin fibroblasts transduced with H3.3-SNAP-HA3 |
| Cell line (*H. sapiens*) | BJ HIRA-HA WT/mSIM(1-5) | This paper | | Human primary foreskin fibroblasts transduced with HIRA-HA WT/mSIM(1-5) |
| Cell line (*H. sapiens*) | BJ EYFP-SP100 isoform A WT or K297R (siRNA resistant) | This paper | | Human primary foreskin fibroblasts transduced with EYFP-SP100 isoform A WT or K297R (siRNA resistant) |
| Cell line (*H. sapiens*) | BJ Myc-DAXX | This paper | | Human primary foreskin fibroblasts transduced with Myc-DAXX |
| Cell line (*H. sapiens*) | IMR90 | ATCC | CCL-186 | Fetal lung primary fibroblasts |
| Cell line (*H. sapiens*) | IMR90 eH3.3i | This paper | | Fetal lung primary fibroblasts transduced with H3.3-SNAP-HA3 |
| Cell line (*H. sapiens*) | IMR90 HIRA-HA WT/K809G | This paper | | Fetal lung primary fibroblasts transduced with HIRA-HA WT/K809G |
| Cell line (*H. sapiens*) | HEK 293T | Intercell, AG *Corpet et al., 2014* | | Immortalized embryonic kidney cells |
| Cell line (*M. musculus*) | MEF PML⁻/⁻ | Gift from Dr. Lallemand-Breitenbach *Sahin et al., 2014* | | Mouse embryonic fibroblasts knocked-out for *PML* |
| Cell line (*M. musculus*) | MEF PML⁻/⁻ Myc-PML1 WT/3 K | This paper | | Mouse embryonic fibroblasts knocked-out for *PML* and transduced with TetOne-Myc-PML1 WT/3 K |
| Antibody | Anti-Actin (Rabbit polyclonal) | Sigma-Aldrich | A2066 | WB 1:1000 |
| Antibody | Anti-panH3 (Rabbit polyclonal) | Abcam | ab1791 | WB 1:5000 ChIP: 2 µg |
| Antibody | Anti-H3.3 (Rabbit monoclonal) | Diagenode | C15210011 | WB 1:1000 ChIP: 2 µg |
| Antibody | Anti-HA (Rabbit polyclonal) | Abcam | ab9110 | IF 1:1000 WB 1:1000 |
| Antibody | Anti-HIRA #01 (Mouse monoclonal, clone WC119) | Active Motif | 3558 | IF 1:500 WB 1:1000 |
| Antibody | Anti-HIRA #02 (Mouse monoclonal, clone WC119) | Millipore | 04–1488 | IF 1:500 WB 1:1000 |
| Antibody | Anti-HIRA #03 (Rabbit polyclonal) | Abcam | ab20655 | For mouse IF 1:100 |
| Antibody | Anti-6xHis #01 (Mouse monoclonal, clone 3D5) | Clontech | 631212 | IF 1:1000 |
| Antibody | Anti-6xHis #02 (Rabbit polyclonal) | Bethyl | A190-114A | IF 1:10000 |

*Appendix 1 Continued on next page*

*Appendix 1 Continued*

| Reagent type (species) or resource | Designation | Source or reference | Identifiers | Additional information |
|---|---|---|---|---|
| Antibody | Anti-c-Myc #01 (Mouse monoclonal, clone 9E10) | Santa Cruz | sc-40 | WB 1:1000 |
| Antibody | Anti-c-Myc #02 (Rabbit polyclonal) | Abcam | ab9106 | IF 1:1000 |
| Antibody | Anti-PML #01 (Mouse monoclonal, clone PG-M3) | Santa Cruz | sc-966 | IF 1:200 |
| Antibody | Anti-PML #02 (Rabbit polyclonal) | Santa Cruz | sc-5621 | IF 1:200 WB 1:1000 |
| Antibody | Anti-PML #03 (Rabbit polyclonal) | Sigma | PLA0172 | IF 1:5000 WB 1:1000 |
| Antibody | Anti-PML #04 (Mouse monoclonal, clone 36.1–104) | Millipore | MAB3738 | For mouse IF 1:100 |
| Antibody | Anti-SP100 (Rabbit polyclonal antiserum) | Kind gift from Dr. Thomas M. Sternsdorf | GH3 | IF: 1:100 WB: 1:1000 |
| Antibody | Anti-SUMO-1 (Rabbit monoclonal, clone Y299) | Abcam | ab32058 | WB 1:1000 |
| Antibody | Anti-SUMO-2/3 (Rabbit polyclonal) | Abcam | ab3742 | WB 1:1000 |
| Antibody | Anti-αTubulin (Mouse monoclonal, clone DM1A) | Sigma | T6199 | WB 1:10000 |
| Recombinant DNA reagent | pLentiN (plasmid) | Addgene | #37444 | Lentiviral plasmid |
| Recombinant DNA reagent | pLVX-TetOne (plasmid) | Clontech | 631849 | Doxycyclin inducible lentiviral plasmid |
| Recombinant DNA reagent | pcDNA5-Tat-S1S2D5-Flag-His (plasmid) | Gift from Dr. Hughes *Hughes et al., 2017* | | |
| Recombinant DNA reagent | pLVX-TetOne-Tat-S1S2D5-Flag-His (plasmid) | This paper | | PCR on pcDNA5-Tat-S1S2D5-Flag-His and cloning in lentiviral pLVX-TetOne plasmid |
| Recombinant DNA reagent | pLentiN-HIRA-HA WT (plasmid) | This paper | | RT-PCR on HeLa cells and cloning in lentiviral pLentiN plasmid |
| Recombinant DNA reagent | pLentiN-HIRA-HA WT (plasmid) | This paper | | RT-PCR on HeLa cells and cloning in lentiviral pLentiN plasmid |
| Recombinant DNA reagent | pLentiN-HIRA-HA mSIM(1-5) (plasmid) | This paper | | Site-directed mutagenesis on pLentiN-HIRA-HA WT plasmid. Mutations: mSIM1=V124 G/S125G; mSIM2=V225 G/L226G; mSIM3=L320 G/L321G; mSIM4=V805 G/V806G; mSIM5=V978 G/V979G |
| Recombinant DNA reagent | pLentiN-HIRA-HA K809G (plasmid) | This paper | | Site-directed mutagenesis K809G on pLentiN HIRA-HA WT plasmid |
| Recombinant DNA reagent | pLNGY-PML1 WT (plasmid) | Gift from Dr. Everett *Cuchet et al., 2011* | | |
| Recombinant DNA reagent | pLNGY-PML1.KKK (plasmid) | Gift from Dr. Everett *Cuchet et al., 2011* | | 3 main SUMOylation sites mutated PML1 |

*Appendix 1 Continued on next page*

*Appendix 1 Continued*

| Reagent type (species) or resource | Designation | Source or reference | Identifiers | Additional information |
|---|---|---|---|---|
| Recombinant DNA reagent | pLVX-PML1 WT (plasmid) | This paper | | PCR on pLNGY-PML1 WT and cloning in lentiviral pLVX-TetOne plasmid |
| Recombinant DNA reagent | pLVX-PML1.KKK (plasmid) | This paper | | PCR on pLNGY-PML1.KKK and cloning in lentiviral pLVX-TetOne plasmid |
| Recombinant DNA reagent | pLNGY-EYFP-S100 isoform A WT (plasmid) | Gift from Dr. Everett ***Cuchet-Lourenço et al., 2011*** | | |
| Recombinant DNA reagent | pLNGY-EYFP-S100 isoform A K297R (plasmid) | Gift from Dr. Everett ***Cuchet-Lourenço et al., 2011*** | | Main SUMOylation site K297 mutated on SP100 |
| Recombinant DNA reagent | pLVX-EYFP-SP100 isoform A WT (plasmid) | This paper | | PCR on pLNGY-EYFP-S100 isoform A WT and cloning in lentiviral pLVX-TetOne plasmid |
| Recombinant DNA reagent | pLVX-EYFP-SP100 isoform A K297R (plasmid) | This paper | | PCR on pLNGY-EYFP-S100 isoform A K297R and cloning in lentiviral pLVX-TetOne plasmid |
| Recombinant DNA reagent | pBABE-H3.3-SNAP-HA3 (plasmid) | Gift from Dr. Jansen | | |
| Recombinant DNA reagent | pLVX-TetOne-H3.3-SNAP-HA3 (plasmid) | This paper | | PCR on pBABE-H3.3-SNAP-HA3 and cloning in lentiviral pLVX-TetOne plasmid |
| Recombinant DNA reagent | RP11-438J1 (BAC) | Gift from Dr. Schluth-Bolard | | BAC used for FISH probe against *GHRL* |
| Recombinant DNA reagent | RP11-185E17 (BAC) | RP11-185E17 | https://bacpacresources.org/ | BAC used for FISH probe against *PML* |
| Recombinant DNA reagent | RP11-120C17 (BAC) | RP11-120C17 | https://bacpacresources.org/ | BAC used for FISH probe against *MX1* |
| Recombinant DNA reagent | RP11-134B23 (BAC) | RP11-134B23 | https://bacpacresources.org/ | BAC used for FISH probe against *OAS1* |
| Sequence-based reagent | siHIRA | ***Ray-Gallet et al., 2011*** | siRNA | sequence 5′GGAUAACACUGUCGUCAUCdTdT |
| Sequence-based reagent | siLuc | ***Adam et al., 2013*** | siRNA | sequence 5′CGUACGCGGAAUACUUCGAdTdT |
| Sequence-based reagent | siPML | ***Everett et al., 2006*** | siRNA | sequence 5′AGATGCAGCTGTATCCAAGdTdT |
| Sequence-based reagent | siSP100 | ***Everett et al., 2008*** | siRNA | sequence 5'GUGAGCCUGUGAUCAAUAAdTdT |
| Sequence-based reagent | siSUMO-1 | ***Lallemand-Breitenbach et al., 2008*** | siRNA | sequence 5′GGACAGGAUAGCAGUGAGAdTdT |
| Sequence-based reagent | siSUMO-2/3 | ***Yao et al., 2011*** | siRNA | sequence 5′GUCAAUGAGGCAGAUCAGAdTdT |
| Sequence-based reagent | H3.3-ChIP-cluster3-Chr1-F (Enh1) | ***Pchelintsev et al., 2013*** | CHIP qPCR primer | 5′GCCACTTGCCAATGTTTCTC |
| Sequence-based reagent | H3.3-ChIP-cluster3-Chr1-R (Enh1) | ***Pchelintsev et al., 2013*** | CHIP qPCR primer | 5′TGGCCCCATGTAGTGAAAAG |
| Sequence-based reagent | ChIP-GCH1-TES-F | this paper | CHIP qPCR primer | 5'TCTGGTCCCGGTTTCCTTTG |
| Sequence-based reagent | ChIP-GCH1-TES-R | this paper | CHIP qPCR primer | 5'TTTAATTTGGCCCACGCTGC |

*Appendix 1 Continued on next page*

*Appendix 1 Continued*

| Reagent type (species) or resource | Designation | Source or reference | Identifiers | Additional information |
|---|---|---|---|---|
| Sequence-based reagent | ChIP-ISG54-TSS-F | This paper | CHIP qPCR primer | 5'GCAGGAAGTGGGGTTTGCTA |
| Sequence-based reagent | ChIP-ISG54-TSS-R | This paper | CHIP qPCR primer | 5'GAGGGATGTTTCATCGGCCT |
| Sequence-based reagent | ChIP-ISG54-mid-F | This paper | CHIP qPCR primer | 5'ATGTAACTAACCCCAGGTGCG |
| Sequence-based reagent | ChIP-ISG54-mid-R | This paper | CHIP qPCR primer | 5'TGCTTCCCACTCCCATTTTGA |
| Sequence-based reagent | ChIP-ISG54-TES-F | This paper | CHIP qPCR primer | 5'AGTCTGGAAGCCTCATCCCT |
| Sequence-based reagent | ChIP-ISG54-TES-R | This paper | CHIP qPCR primer | 5'CCTAGTGGGCACCACATCTC |
| Sequence-based reagent | ChIP-MX1-TSS-F | *Cheon et al., 2013* | CHIP qPCR primer | 5'GCCCTCTCTTCTTCCAGGCAAC |
| Sequence-based reagent | ChIP-MX1-TSS-R | *Cheon et al., 2013* | CHIP qPCR primer | 5'GGGACAGGCATCAACAAAGC |
| Sequence-based reagent | ChIP-MX1-mid-F | This paper | CHIP qPCR primer | 5'TCTACGCTCTGGGGACATCA |
| Sequence-based reagent | ChIP-MX1-mid-R | This paper | CHIP qPCR primer | 5'GAACCAAACCCACCACCAGA |
| Sequence-based reagent | ChIP-MX1-TES-F | This paper | CHIP qPCR primer | 5'CTCCCGTGAACTGTTCTTTCCT |
| Sequence-based reagent | ChIP-MX1-TES-R | This paper | CHIP qPCR primer | 5'GCTGTAGGTGTCCTTGTCCT |
| Sequence-based reagent | ChIP-MX2-TES-F | This paper | CHIP qPCR primer | 5'ACCACTCCAGCAAACCCTTC |
| Sequence-based reagent | ChIP-MX2-TES-R | This paper | CHIP qPCR primer | 5'AATGGGATCTGGTTGGCGAG |
| Sequence-based reagent | ChIP-OAS1-TSS-F | This paper | CHIP qPCR primer | 5'ACCACAGACAACTGTGAAAGG |
| Sequence-based reagent | ChIP-OAS1-TSS-R | This paper | CHIP qPCR primer | 5'GTCCTTTAGCCAGCAACAAGC |
| Sequence-based reagent | ChIP-OAS1-mid-F | This paper | CHIP qPCR primer | 5'GCAGCACGTTGGGAGATAGA |
| Sequence-based reagent | ChIP-OAS1-mid-R | This paper | CHIP qPCR primer | 5'TTCTCCTGATGTGGCAAGGG |
| Sequence-based reagent | ChIP-OAS1-TES-F | This paper | CHIP qPCR primer | 5'CTTGTCACATCCCCACCTCTC |
| Sequence-based reagent | ChIP-OAS1-TES-R | This paper | CHIP qPCR primer | 5'GTCCTTTGCCCCTGTTTAGC |
| Sequence-based reagent | GAPDH-F | This paper | RT qPCR primer | 5'GAGTCAACGGATTTGGTCG |
| Sequence-based reagent | GAPDH-R | This paper | RT qPCR primer | 5'TTGATTTTGGAGGGATCTCG |
| Sequence-based reagent | H3F3A-F | This paper | RT qPCR primer | 5'CCAGGAAGCAACTGGCTACA |
| Sequence-based reagent | H3F3A-R | This paper | RT qPCR primer | 5'ACCAGGCCTGTAACGATGAG |

*Appendix 1 Continued on next page*

*Appendix 1 Continued*

| Reagent type (species) or resource | Designation | Source or reference | Identifiers | Additional information |
|---|---|---|---|---|
| Sequence-based reagent | HIRA-F | This paper | RT qPCR primer | 5'AGGACTCTCGTCTCATGCCT |
| Sequence-based reagent | HIRA-R | This paper | RT qPCR primer | 5'CAGCTTCAGTGCAAGTGCT |
| Sequence-based reagent | ISG15-F | This paper | RT qPCR primer | 5'GGTGGACAAATGCGACGAAC |
| Sequence-based reagent | ISG15-R | This paper | RT qPCR primer | 5'TCGAAGGTCAGCCAGAACAG |
| Sequence-based reagent | ISG54-F | This paper | RT qPCR primer | 5'TGAAAGAGCGAAGGTGTGCT |
| Sequence-based reagent | ISG54-R | This paper | RT qPCR primer | 5'CTCAGAGGGTCAATGGCGTT |
| Sequence-based reagent | MX1-F | This paper | RT qPCR primer | 5'GGAGGCACTGTCAGGAGTTG |
| Sequence-based reagent | MX1-R | This paper | RT qPCR primer | 5'TCCTGGTAACTGACCTTGCC |
| Sequence-based reagent | OAS1-F | This paper | RT qPCR primer | 5'AGCTGGAAGCCTGTCAAAGA |
| Sequence-based reagent | OAS1-R | This paper | RT qPCR primer | 5'AGGTTTATAGCCGCCAGTCA |
| Sequence-based reagent | PML-F | This paper | RT qPCR primer | 5'CAGGGACCCTATTGACGTTG |
| Sequence-based reagent | PML-R | This paper | RT qPCR primer | 5'ATGGAGAAGGCGTACACTGG |
| Sequence-based reagent | SP100-F | This paper | RT qPCR primer | 5'CACTGACGTTGATGAGCCCT |
| Sequence-based reagent | SP100-R | This paper | RT qPCR primer | 5'AATCTGGGGTCGTGAGCAAG |
| Peptide, recombinant protein | IFNβ | Peprotech | 300-02BC | Human Final concentration: 100 or 1000 U/mL |
| Peptide, recombinant protein | IFNα | PBL assay science | 12105–1 | Mouse Final concentration: 1000 U/mL |
| Peptide, recombinant protein | IL-6 | Peprotech | 200–06 | Human Final concentration: 200 ng/mL |
| Peptide, recombinant protein | IL-8 | Peprotech | 200–08 | Human Final concentration: 200 ng/mL |
| Peptide, recombinant protein | TNF-α | Invivogen | rcyc-htnfa | Human Final concentration: 100 ng/mL |
| Chemical compound, drug | Blasticidin | Invivogen | ant-bl | Final concentration: 5 µg/mL |
| Chemical compound, drug | Doxycycline | Sigma Aldrich | D9891 | Final concentration: 100 ng/mL |
| Chemical compound, drug | EdU | Invitrogen | C10338 | Final concentration: 10 µM |
| Chemical compound, drug | PolyI:C | Invivogen | trlr-pic | Final concentration: 10 µg/mL |
| Chemical compound, drug | Puromycin | Invivogen | ant-pr | Final concentration: 1 µg/mL |

*Appendix 1 Continued on next page*

*Appendix 1 Continued*

| Reagent type (species) or resource | Designation | Source or reference | Identifiers | Additional information |
|---|---|---|---|---|
| Chemical compound, drug | Ruxolitinib | Invivogen | trlr-rux | Final concentration: 2 μM |
| Commercial assay or kit | Duolink In Situ Red Starter Kit Mouse/Rabbit | Sigma Aldrich | DUO092101 | For PLA |
| Software, algorithm | Homer | http://homer.ucsd.edu/homer/download.html | | ChIP seq analysis |
| Software, algorithm | MACS2 | *Zhang et al., 2008* | | ChIP seq analysis |
| Software, algorithm | Bowtie2 | *Langmead and Salzberg, 2012* | | ChIP seq analysis |
| Software, algorithm | DeepTools suite | https://deeptools.readthedocs.io/en/develop/ | | ChIP seq visualization |
| Software, algorithm | enrichR | *Kuleshov et al., 2016* | | GO analysis |
| Software, algorithm | IGV | https://software.broadinstitute.org/software/igv/ | | Genome browser |
| Software, algorithm | Fiji | https://imagej.net/imagej-wiki-static/Fiji/Downloads | | Image analysis |
| Software, algorithm | Photoshop | https://www.adobe.com/products/photoshop.html | | Image analysis |
| Software, algorithm | GraphPad Prism 6 | https://www.graphpad.com/scientific-software/prism/ | | Statistical analysis and graphics |
| Software, algorithm | Biorender | biorender.com | | figure creation |

