## [Editor Report]

This study nicely dissects unexpected crosstalk between PML nuclear bodies and the HIRA member of the H3.3 histone chaperone complex upon inflammatory stress. The work raises interesting perspectives on how availability of HIRA could be regulated by PML Nuclear Bodies for histone deposition, and transcriptional regulation of key interferon-stimulated genes. Overall, this is an important step forward in unveiling which epigenetic pathways might regulate immune responses.

---

## [Decision Letter]

**Decision letter after peer review:**

Thank you for submitting your article "Interplay between PML NBs and HIRA for H3.3 dynamics following type I interferon stimulus" for consideration by *eLife*. Your article has been reviewed by 2 peer reviewers, one of whom is a member of our Board of Reviewing Editors, and the evaluation has been overseen by Jessica Tyler as the Senior Editor. The reviewers have opted to remain anonymous.

The reviewers have discussed their reviews with one another, and the Reviewing Editor has drafted this to help you prepare a revised submission. Please note that the revision will go back to the reviewers for comments on its suitability for publication.

Essential revisions:

The central issue with the manuscript is that both reviewers found the premise interesting, but found the evidence presented somewhat lacking in strength. I would invite you to submit a revision that directly addresses these key points made by both reviewers.

1. Both reviewers found the claims of valency puzzling and lacking in evidence – I would suggest either buttressing these claims rigorously or perhaps toning down/removing these claims entirely.

2. Both reviewers suggest tightening the manuscript to focus on H3.3 chaperone-PML interactions. For e.g. reviewers noted that the evidence for HIRA interactions with PML was a bit weak, and suggested examining DAXX (another H3.3 chaperone). I concur.

3. Reviewers asked whether OE of HIRA and its potential deposition into PML NBs is SUMO-dependent or SUMO-independent. Overexpressing the SIM mutants from Figure 3F would address this question. In addition, the link between the proposed HIRA being stored at PML NBs could be strengthened by overexpressing HIRA and see at both short and late time points whether H3.3 is enriched on ISG genes.

4. Both reviewers commented on the lack of controls and quantification for all WBs- these will be necessary for reviewers to assess the strength of claims which rely on WB measurements.

5. Reviewer 2 had several specific comments on data that require clarifications, in some cases textual revisions will suffice (e.g. valency), in others, you may wish to provide additional evidence to buttress your major claims. E.g. for the PML mutants, ectopic localization is decreased after interferon treatment. These puzzling/contrary results could benefit from additional experiments, possibly looking at other partners in the HIRA complex.

*Reviewer #2 (Recommendations for the authors):*

The manuscript should be improved with a more straightforward purpose, focused on HIRA/PML interactions and H3.3 loading upon interferon stimulation. The current presentation is a bit puzzling first focused on what the authors call "condensation and valency"; then on the regulation by PML – but not HIRA – of ISG expression upon IFN β; then focus again on the role of HIRA and PML in H3.3 loading at ISG genes; and finally, back to HIRA localization at PML NBs upon H3.3 overexpression and acetylation inhibition. Many of these observations are interesting but not related to each other.

p5, according to my knowledge, there is a confusion between "valency" and "concentration" in the LLPS process. Valency refers to the number of interacting domains in a given molecule, high valency and high affinity controlling phase separation of molecules at low concentration.

First, the authors should rephrase their sentence concerning PML NBs, SUMOs and SIM (p5). In the publication "Banani et al. 2016", the valency in droplet formation was investigated by changing the number of repeats of SUMO or SIMs in chimeric fluorescent proteins, mixed at various concentrations in vitro. In this publication, PML was used to support their conclusion on the role of stoichiometry in cells, showing that their fluorescent chimeric poly-SIM or poly-SUMO proteins were differentially recruited depending on the availability of SUMO moieties in their overexpressed GFP-PML. In cells, and in contrast to the phase diagram established in vitro, concentration of GFP-PML was not changed (nor the number of conjugated SUMO) and the co-condensation of GFP-polySIM with RFP-polySUMO chimeras was not assessed. For clarity of the reading, the authors should separate, using two sentences, the role of multiple SIM and SUMO in LLPS – depending on both valency, affinity, and concentration (Banani 2016) -, from the role PML sumoylation to recruit SIM-proteins in cells (Sahin 2014).

Then, in figure 1, the authors studied HIRA localization at PML NBs when PML protein amount was increased by IFN-I pathway stimulations (known to activate PML transcription). They did not evaluate "PML valency", but rather PML concentration increase. Indeed, poly(I:C), TNF or INF β treatments increased both HIRA at PML NBs and PML protein amount. PML protein amount was not affected by IL6 or 10 exposure, which had no effect on HIRA localization (Figure 1C and Figure 1-supplementary1 B). More importantly, both unsumoylated and sumoylated PML proteins were increased upon IFN β or poly(I:C) treatment, but the fraction of sumoylated PML relative to total PML proteins did not change compared to untreated cells (Figure 1C).

Assessing PML transcript level upon treatments would also help in better characterizing the change in PML amount indicated by WB (quantification of WB?).

In addition, in figures 2 and 3, neither the number of SUMOylation sites on PML required for HIRA recruitment, nor its polySUMOylation were not assessed.

In figure 3F, the authors identified a putative SIM2, but not SIM4 nor 5, as required for HIRA localization at PML NBs. This supports a specific role of HIRA SIM2 sequence rather than a general effect of valency (number of interacting domains in HIRA). Thus, the authors should rephrase the title and content of this paragraph to indicate that HIRA accumulation at PML nuclear bodies correlates with increase in PML concentration, rather than valency.

How do the authors explain that ectopic PML and PML 3K mutant decreased after interferon treatment? The effect is huge and requires at least an explanation. Is it an inhibition of the transcriptional induction by doxycycline by INF β signaling?

How are localized the other members of the HIRA complex upon interferon β and in Pml-/- MEFs expressing PMLWT or 3K mutant?

The authors conclude from Figures 1 and 2 that HIRA recruitment at PML NBs upon IFN β (or TNF, poly(I:C)) correlates with PML amount and depends on sumoylation. In Figure 1—figure supplementary 1D, the amount of endogenous HIRA looks higher upon IFN β, as in Figure3D or in HIRA IF in 3E. The authors should assess whether this increase is systematic. In particular, HIRA transcription level should be quantified in IFN β-treated compared to untreated cells, and quantification of HIRA proteins relative to loading controls should be shown.

In addition, it is unclear to me why addition of IFN β is required for HIRA localization at PML NBs in pml-/- MEFs transduced with Dox-inducible myc-PML (Figure 2-supplementary 1D; Figure 3D). IFN β signaling cannot increase PML transcription in those cells. How could the author explain here the need for IFNβ? Again, is it relative to any HIRA increase by IFNβ? Could HIRA increase contribute to HIRA localization at PML NBs upon IFN β (as suggested by figure 6)?

Alternatively, this could be mediated by increase in SP100. Indeed, SP100 is, as PML, a wellknown interferon-upregulated gene. The authors confirmed SP100 is required for HIRA localization at PML NBs upon INF β (Figure 3-supplementary 1). They should also assess whether the SP100 increase could explain the IFN effect on PML ectopically expressed.

There are several SP100 isoforms, and all should be also induced by interferon signaling. Yet, IFN β did not increase endogenous SP100 proteins in the experiment shown in Figure 3—figure supplementary1 B. A larger crop of the Western blot analysis would allow to check both the effect of the SiRNA and interferon on the SP100A, B, C and HMG isoforms. To my view, the authors could investigate further this requirement of SP100 for the mechanism of HIRA trafficking and localization at PML NBs and, especially regarding the role of SUMO and SIM. It has been shown that SP100 is recruited by sumoylated PML at PML bodies and may be itself sumoylated there, at least upon stress. Is the sumoylation site of SP100 required for HIRA recruitment at PML NBs?

Then, the authors assessed ISG expression, HIRA-containing PML NBs localization relative to ISG loci, and then focused on H3.3 loading at transcriptional-end site. Figure 4A shows that HIRA is not involved in IFN-induced ISG expression, which only requires PML, as an early process. This looks unrelated to the focus of the manuscript (HIRA/PML/chromatin status). The authors should move Figure 4A to supplementary figures. The text should also to be shortened to maintain a clear focus on the crosss talk.

In addition, baseline levels of ISG transcripts upon HIRA and PML knock down should be shown (in supplementary), comparing BJ cells with siHIRA or siPML to those transfected with the control siLuc in absence of IFNβ treatment.

The authors highlighted the kinetic aspects of the H3.3 increase at the TES of ISGs from 6 to 48h after IFN treatment, while the maximal expression of ISGs was obtained at 6h. If PML and HIRA sustain H3.3 at TES, siPML and siHIRA should impair this kinetic. The authors should also assess H3.3 fold change at some ISG TES with time.

Although of interest, the data presented in Figure 6 seems quite preliminary to conclude on a HIRA buffering role of PML NBs, and looks more related to the mechanism by which HIRA localization is controlled. SIM is usually a b-strand-like conformation and interacts with a groove of SUMO. Is the SIM2 sequence still accessible when HIRA is bound to H3.3 or in the whole HIRA chaperone complex? TSA could alter acetylation more broadly, which was also proposed to regulate SUMO/SIM interactions.

---

## [Author Response]

Essential revisions:The central issue with the manuscript is that both reviewers found the premise interesting, but found the evidence presented somewhat lacking in strength. I would invite you to submit a revision that directly addresses these key points made by both reviewers.1. Both reviewers found the claims of valency puzzling and lacking in evidence – I would suggest either buttressing these claims rigorously or perhaps toning down/removing these claims entirely.

We have removed the claims on PML valency. We now underscore the link between the increase in PML and SP100 protein levels and the accumulation of HIRA in PML NBs, without lingering on the valency aspects which was not the focus of our paper.

2. Both reviewers suggest tightening the manuscript to focus on H3.3 chaperone-PML interactions. For e.g. reviewers noted that the evidence for HIRA interactions with PML was a bit weak, and suggested examining DAXX (another H3.3 chaperone). I concur.

In the first part of the manuscript, we have now explored in more details the interplay between HIRA and DAXX regarding their localization in PML NBs. Specifically, we show that DAXX depletion triggers a moderate, but significant increase in HIRA accumulation in PML NBs, while DAXX overexpression completely abrogates HIRA accumulation in PML NBs upon IFN-I treatment.

We believe these new data support our claims that HIRA localization in PML NBs is dependent on the availability of free binding sites and that both PML and SP100 concentrations as well as DAXX and H3.3 levels regulate HIRA accumulation in PML NBs.

3. Reviewers asked whether OE of HIRA and its potential deposition into PML NBs is SUMO-dependent or SUMO-independent. Overexpressing the SIM mutants from Figure 3F would address this question. In addition, the link between the proposed HIRA being stored at PML NBs could be strengthened by overexpressing HIRA and see at both short and late time points whether H3.3 is enriched on ISG genes.

We have now included in Figure 3-supplement Figure 1C data with overexpression of the HIRA mSIM mutant without IFN-I treatment. This mutant shows impaired accumulation of HIRA in PML NBs with or without IFN-I (Figure 3F and Figure 3-supplement Figure 1C), suggesting that accumulation of HIRA in PML NBs is dependent on SUMO-SIM interactions regardless of the IFN-I treatment.

In addition, we have now included ChIP-qPCR data on H3.3 enrichment on ISGs after HIRA overexpression (Figure 6-supplement figure 2A-B). We see a moderate increase in H3.3 enrichment when HIRA is overexpressed. However, as stated below, we believe that HIRA accumulation in PML NBs and its role in H3.3 deposition on ISGs are two independent events. Thus, overexpression of HIRA triggers both HIRA accumulation in PML NBs as part of a buffering mechanism and can lead to increase deposition of H3.3 on ISGs due to the increase of the nucleoplasmic pool of HIRA available.

4. Both reviewers commented on the lack of controls and quantification for all WBs- these will be necessary for reviewers to assess the strength of claims which rely on WB measurements.

We have now included quantification of all western blots below each depicted gel.

5. Reviewer 2 had several specific comments on data that require clarifications, in some cases textual revisions will suffice (e.g. valency), in others, you may wish to provide additional evidence to buttress your major claims. E.g. for the PML mutants, ectopic localization is decreased after interferon treatment. These puzzling/contrary results could benefit from additional experiments, possibly looking at other partners in the HIRA complex.

We have carefully addressed all reviewers' comments requiring clarifications, e.g. on valency issues. Concerning the localization of the ectopic PML WT and mutant, their localization is not decreased after IFN treatment, but it is just their expression that is lower than in absence of mIFNa. Nevertheless, the expression of these ectopic PML is sufficient to be seen on a single cell basis by immunofluorescence, and thus allows us to conclude that the PML 3K mutant does not rescue HIRA localization in PML NBs.

We believe that our pLVX-TetOne inducible plasmid (used either for Myc-PML or H3.3-HA ectopic expression) is less induced by doxycyclin when IFN-I is added in the medium. We haven't found the reason for this observation but speculate that the massive increase in transcription after IFN-I addition might reduce the doxycyclin-inducible expression of the plasmid due to a limitation in the number of RNA polymerases available. In any case, this effect does not prevent us from drawing firm conclusions since we compare the rescue obtained with PML WT or PML 3K mutant in similar conditions (+dox +mIFNa) (Figures 3D-E). We already clearly state in the text that "Despite a diminution in the amount of the ectopic PML proteins following addition of mouse IFNa (Figure 3D), the wild type PML rescued HIRA accumulation in ectopically formed PML NBs unlike the PML 3K (Figure 3E)."

Concerning the other partners of the HIRA complex, it is already shown that they are all present in PML NBs together with HIRA (Cohen et al., 2018) but we also provide an IF (Author response image 1) of all members of the HIRA complex present in PML NBs upon IFNβ treatment in human primary fibroblasts. Unfortunately, we were unable to show the whole HIRA complex in mouse embryonic fibroblasts, due to issues with antibodies not working in our hands in mouse cells.

**Author response image 1. sa2fig1:** Members of the HIRA complex accumulate in PML NBs following IFN-I treatment. Fluorescence microscopy visualization of ASF1 (green), UBN1 (green) or CABIN1 (green) together with PML (red) in BJ cells treated with IFNβ at 1000U/mL for 24h (+IFNβ) or left untreated (NT). Cell nuclei are visualized by DAPI staining (grey). Scale bar represents 10 μm.

Reviewer #2 (Recommendations for the authors):The manuscript should be improved with a more straightforward purpose, focused on HIRA/PML interactions and H3.3 loading upon interferon stimulation. The current presentation is a bit puzzling first focused on what the authors call "condensation and valency"; then on the regulation by PML – but not HIRA – of ISG expression upon IFN β; then focus again on the role of HIRA and PML in H3.3 loading at ISG genes; and finally, back to HIRA localization at PML NBs upon H3.3 overexpression and acetylation inhibition. Many of these observations are interesting but not related to each other.

We thank the reviewer for pointing out the absence of focus in the manuscript. As stated above, we have now remodeled the manuscript in a more straightforward manner. We now clearly convey two main conclusions regarding (1) the mechanistic accumulation of HIRA in PML NBs which could act as buffering places for HIRA and (2) the independent role of PML in regulating ISGs transcription and thus indirectly regulating HIRA loading on ISGs, with both proteins thus impacting on H3.3 deposition. These two conclusions are indeed independent of each other, but we believe that it is very interesting that they are put together in a same article exactly to put forward that these are two independent roles of PML/PML NBs in regulating HIRA dynamics.

A long-standing question in the field is indeed to understand the role of HIRA accumulation in PML NBs and whether PML NBs could play a direct role in providing HIRA on neighboring loci. We believe that our data now prove that these are two independent functions of PML NBs. HIRA can accumulate in PML NBs upon IFN-I treatment, and independently be loaded on ISGs to regulate H3.3 deposition. Indeed, the role of PML in ISGs transcription indirectly regulates HIRA loading on ISGs upon IFN-I. In addition, our data using SP100 knock-down which totally abrogates HIRA localization in PML NBs does not prevent HIRA-mediated H3.3 deposition on ISGs confirming that these are two independent events. HIRA mediated H3.3-deposition is highly linked to the transcriptional status of ISGs since upon PML knock-down, there is less ISGs transcription, less HIRA loading on ISGs and less H3.3 deposition. On the contrary, upon SP100 knock-down, there is an increase in ISGs transcription, mirrored by an increase in H3.3 deposition mediated by HIRA, as shown with our new ChIP-qPCR data on H3.3 in the double SP100+HIRA knock-down.

p5, according to my knowledge, there is a confusion between "valency" and "concentration" in the LLPS process. Valency refers to the number of interacting domains in a given molecule, high valency and high affinity controlling phase separation of molecules at low concentration.First, the authors should rephrase their sentence concerning PML NBs, SUMOs and SIM (p5). In the publication "Banani et al. 2016", the valency in droplet formation was investigated by changing the number of repeats of SUMO or SIMs in chimeric fluorescent proteins, mixed at various concentrations in vitro. In this publication, PML was used to support their conclusion on the role of stoichiometry in cells, showing that their fluorescent chimeric poly-SIM or poly-SUMO proteins were differentially recruited depending on the availability of SUMO moieties in their overexpressed GFP-PML. In cells, and in contrast to the phase diagram established in vitro, concentration of GFP-PML was not changed (nor the number of conjugated SUMO) and the co-condensation of GFP-polySIM with RFP-polySUMO chimeras was not assessed. For clarity of the reading, the authors should separate, using two sentences, the role of multiple SIM and SUMO in LLPS – depending on both valency, affinity, and concentration (Banani 2016) -, from the role PML sumoylation to recruit SIM-proteins in cells (Sahin 2014).

We thank the reviewer for this very useful comment and have now modified the text accordingly.

Then, in Figure 1, the authors studied HIRA localization at PML NBs when PML protein amount was increased by IFN-I pathway stimulations (known to activate PML transcription). They did not evaluate "PML valency", but rather PML concentration increase. Indeed, poly(I:C), TNF or INF β treatments increased both HIRA at PML NBs and PML protein amount. PML protein amount was not affected by IL6 or 10 exposure, which had no effect on HIRA localization (Figure 1C and Figure 1-supplementary1 B). More importantly, both unsumoylated and sumoylated PML proteins were increased upon IFN β or poly(I:C) treatment, but the fraction of sumoylated PML relative to total PML proteins did not change compared to untreated cells (Figure 1C).

We agree with the reviewer that we did not assess PML 'valency' per se but only its protein and SUMOylation levels. As stated above, we have now removed all claims on 'valency' and only focus on the increase in PML and SP100 protein levels. We have added a WB showing SP100 levels in Figure 1D.

Assessing PML transcript level upon treatments would also help in better characterizing the change in PML amount indicated by WB (quantification of WB?).

We have now included quantification of all WBs. We have also added in Figure 1 —figure supplement 1B the transcripts levels for PML and SP100 upon IFN, polyIC and TNFa treatment.

In addition, in Figures 2 and 3, neither the number of SUMOylation sites on PML required for HIRA recruitment, nor its polySUMOylation were not assessed.In Figure 3F, the authors identified a putative SIM2, but not SIM4 nor 5, as required for HIRA localization at PML NBs. This supports a specific role of HIRA SIM2 sequence rather than a general effect of valency (number of interacting domains in HIRA). Thus, the authors should rephrase the title and content of this paragraph to indicate that HIRA accumulation at PML nuclear bodies correlates with increase in PML concentration, rather than valency.

We have now rephrased the title and content of this paragraph according to the relevant suggestions of the reviewer.

How do the authors explain that ectopic PML and PML 3K mutant decreased after interferon treatment? The effect is huge and requires at least an explanation. Is it an inhibition of the transcriptional induction by doxycycline by INF β signaling?

As stated above, we believe that our pLVX-TetOne inducible plasmid (used either for Myc-PML or H3.3-HA ectopic expression) is less induced by doxycyclin when IFN-I is added in the medium. We haven't found the reason for this observation but speculate that the massive increase in transcription after IFN-I addition might reduce the doxycyclin-inducible expression of the plasmid due to a limitation in the number of RNA polymerases available. In any case, this effect does not prevent us from drawing firm conclusions since we compare the rescue obtained with PML WT or PML 3K mutant in similar conditions (+dox +mIFNa) (Figures 3D-E).

We have now added a small sentence in the text to better explain this effect: "Despite a recurrent diminution in the amount of the ectopic PML proteins following addition of mouse IFNa, possibly because of a lower efficiency of the doxycyline induction of ectopic PML transcription in IFN-I treated cells (Figure 3D), the wild type PML rescued HIRA accumulation in ectopically formed PML NBs unlike the PML 3K (Figure 3E)."

How are localized the other members of the HIRA complex upon interferon β and in Pml-/- MEFs expressing PMLWT or 3K mutant?

We thank the reviewer for this interesting comment. Unfortunately, despite several attempts, we could not find reasonably good antibodies marking mouse Asf1a, Ubn1 or Cabin1 to assess this question. We provide Author response image 1 showing the localization of the three other members of the HIRA complex in human primary fibroblasts (ASF1A, UBN1 and CABIN1) (as already shown in Cohen et al., Plos Pathogens 2018) (HIRA is shown in Figure 1 – supplement figure 1A).

The authors conclude from Figures 1 and 2 that HIRA recruitment at PML NBs upon IFN β (or TNF, poly(I:C)) correlates with PML amount and depends on sumoylation. In Figure 1—figure supplementary 1D, the amount of endogenous HIRA looks higher upon IFN β, as in Figure3D or in HIRA IF in 3E. The authors should assess whether this increase is systematic. In particular, HIRA transcription level should be quantified in IFN β-treated compared to untreated cells, and quantification of HIRA proteins relative to loading controls should be shown.

We thank the reviewer for this suggestion. We have now quantified all WBs and indeed see a slight significant increase in HIRA protein levels (1.34 fold) upon IFNβ (Figure 1 —figure supplement 1C), correlating with a slight increase in HIRA mRNA levels as assessed by RT-qPCR in cells treated or not with IFNβ (Figure 1 —figure supplement 1C). While we do not think that this small increase is responsible for HIRA accumulation in PML NBs upon IFNβ (as compared to the 40 fold increase upon HIRA overexpression which triggers ectopic HIRA accumulation in PML NBs without the need for IFNβ), it might slightly contribute to it. We have now rephrased the text according to this result.

In addition, it is unclear to me why addition of IFN β is required for HIRA localization at PML NBs in pml-/- MEFs transduced with Dox-inducible myc-PML (Figure 2-supplementary 1D; Figure 3D). IFN β signaling cannot increase PML transcription in those cells. How could the author explain here the need for IFNβ? Again, is it relative to any HIRA increase by IFNβ? Could HIRA increase contribute to HIRA localization at PML NBs upon IFN β (as suggested by figure 6)?

In Figure 3 —figure supplementary 1B, we use mouse IFNa (and not IFNβ) to show that HIRA accumulation in PML NBs is conserved in MEFs Pml+/+, and lost in MEFs Pml-/-. We agree with the reviewer that in our rescue system, in principle, mouse IFNa cannot increase transcription of the exogenous PML in these cells. Yet, in order to be in similar conditions to compare with MEFs PML-/-, we used mouse IFNa to show that exogenous PML WT can rescue HIRA accumulation in PML NBs while exogenous PML 3K cannot. HIRA protein levels do no show major changes upon mouse IFNa addition in MEF cells as shown by WB quantification (Figure 3D). We hypothesize that PML overexpression and/or increase in endogenous SP100 upon mouse IFNa addition (as suggested below by the reviewer) is sufficient to induce HIRA accumulation in PML NBs.

Alternatively, this could be mediated by increase in SP100. Indeed, SP100 is, as PML, a wellknown interferon-upregulated gene. The authors confirmed SP100 is required for HIRA localization at PML NBs upon INF β (Figure 3-supplementary 1). They should also assess whether the SP100 increase could explain the IFN effect on PML ectopically expressed.

Unfortunately, we could not assess this since our SP100 antibody did not work in mouse cells.

There are several SP100 isoforms, and all should be also induced by interferon signaling. Yet, IFN β did not increase endogenous SP100 proteins in the experiment shown in Figure 3—figure supplementary1 B. A larger crop of the Western blot analysis would allow to check both the effect of the SiRNA and interferon on the SP100A, B, C and HMG isoforms.

We agree with the reviewer that the WB was not appropriate for SP100. We have now put a new WB (Figure 3 —figure supplement 1B or 1F) clearly showing the strong increase in SP100 protein levels upon addition of IFNβ, consistent with our new WB added in Figure 1D. Our siRNA against SP100 is taken from Everett and Orr, J. Virol 2008 (doi:10.1128/JVI.0230807) and targets all SP100 isoforms.

To my view, the authors could investigate further this requirement of SP100 for the mechanism of HIRA trafficking and localization at PML NBs and, especially regarding the role of SUMO and SIM. It has been shown that SP100 is recruited by sumoylated PML at PML bodies and may be itself sumoylated there, at least upon stress. Is the sumoylation site of SP100 required for HIRA recruitment at PML NBs?

We now provide novel data (Figure 3 —figure supplement 1D-F) analyzing the role of SP100 SUMOylation site in the localization of HIRA in PML NBs. We found that EYFP-SP100 K297R localizes correctly in PML NBs even in absence of endogenous SP100 (Author response image 2), as shown previously (Cuchet-Lurenço et al., 2011). EYFP-SP100 K297R rescued HIRA localization in PML NBs to the same extent as the EYFP-SP100 WT protein (Figure 3 —figure supplement 1D-E). Thus, while SP100 plays an important role in the recruitment of HIRA in PML NBs, its SUMOylation *per se* does not seem to be required for HIRA accumulation in PML NBs. This is not antinomic with the role of HIRA putative SIM for its localization in PML NBs. We hypothesize that HIRA could both be dependent on SP100 levels and on the SUMOylation of other proteins (such as PML) for its accumulation in PML NBs. SP100 could play the role of an adapter protein facilitating the recruitment of HIRA next to PML NBs but its internalization would remain dependent on the SUMOylation of PML.

**Author response image 2. sa2fig2:** EYFP-SP100 localizes in PML NBs in absence of endogenous SP100. Fluorescence microscopy visualization of HIRA (green) together with PML (red) and EYFP-SP100 (pseudocolored in cyan) in BJ cells treated with an siRNA against SP100 for 48h and transduced with a lentivirus expressing EYFP-SP100 WT or K297R for 48h. Cell nuclei are visualized by DAPI staining (grey). Scale bar represents 10 μm.

Then, the authors assessed ISG expression, HIRA-containing PML NBs localization relative to ISG loci, and then focused on H3.3 loading at transcriptional-end site. Figure 4A shows that HIRA is not involved in IFN-induced ISG expression, which only requires PML, as an early process. This looks unrelated to the focus of the manuscript (HIRA/PML/chromatin status). The authors should move Figure 4A to supplementary figures. The text should also to be shortened to maintain a clear focus on the crosss talk.

We have now put Figure 4A in Figure 5 —figure supplement 1C and shortened a bit the manuscript in this section.

In addition, baseline levels of ISG transcripts upon HIRA and PML knock down should be shown (in supplementary), comparing BJ cells with siHIRA or siPML to those transfected with the control siLuc in absence of IFNβ treatment.

We have now added the baseline levels of ISGs transcription in absence of IFNβ treatment in HIRA and PML knock downs in Figure 5 —figure supplement 1C.

The authors highlighted the kinetic aspects of the H3.3 increase at the TES of ISGs from 6 to 48h after IFN treatment, while the maximal expression of ISGs was obtained at 6h. If PML and HIRA sustain H3.3 at TES, siPML and siHIRA should impair this kinetic. The authors should also assess H3.3 fold change at some ISG TES with time.

While surely interesting, we haven't performed this experiment. In terms of H3.3 deposition at ISGs, we decided to focus our novel experiments on the role of HIRA in the increased deposition of H3.3 deposition upon SP100 knock down (new Figure 7A-B).

Although of interest, the data presented in Figure 6 seems quite preliminary to conclude on a HIRA buffering role of PML NBs, and looks more related to the mechanism by which HIRA localization is controlled. SIM is usually a b-strand-like conformation and interacts with a groove of SUMO. Is the SIM2 sequence still accessible when HIRA is bound to H3.3 or in the whole HIRA chaperone complex? TSA could alter acetylation more broadly, which was also proposed to regulate SUMO/SIM interactions.

We agree with the reviewer that Figure 6 is more related to the mechanism by which HIRA is controlled. We have now moved this figure upwards in the manuscript (now figure 4) and have completed our data on the role of H3.3 levels with the role of DAXX in controlling HIRA accumulation in PML NBs. In addition, we have removed the TSA data which was indeed quite preliminary.